# Riemannian Stochastic Interpolants for Amorphous Particle Systems

## Abstract

Modern generative models hold great promise for accelerating diverse tasks involving the simulation of physical systems, but they must be adapted to the specific constraints of each domain. Significant progress has been made for biomolecules and crystalline materials. Here, we address amorphous materials (glasses), which are disordered particle systems lacking atomic periodicity. Sampling equilibrium configurations of glass-forming materials is a notoriously slow and difficult task. This obstacle could be overcome by developing a generative framework capable of producing equilibrium configurations with well-defined likelihoods. In this work, we address this challenge by leveraging an equivariant Riemannian stochastic interpolation framework which combines Riemannian stochastic interpolant and equivariant flow matching. Our method rigorously incorporates periodic boundary conditions and the symmetries of multi-component particle systems, adapting an equivariant graph neural network to operate directly on the torus. Our numerical experiments on model amorphous systems demonstrate that enforcing geometric and symmetry constraints significantly improves generative performance.

## 1 Introduction

Successes of diffusion models, flow matching, and stochastic interpolants in generative modeling have led to scientific applications, particularly for generating microscopic configurations in physical systems. Such physics systems pose unique challenges: data are often scarce, and substantial prior knowledge exists in the form of invariances. Adapting generative models to these constraints has yielded notable successes for biomolecules (Geffner et al., 2025; Lewis et al., 2025) and their ligands (Corso et al., 2022), or for crystalline materials (Miller et al., 2024; Höllmer et al., 2025).

Generative models are often trained to approximate an unknown data distribution and evaluated through visual fidelity or likelihood metrics. By contrast, sampling from a known energy landscape requires methods that (i) efficiently generate configurations with high Boltzmann weight and (ii) provide tractable likelihoods or importance weights so that thermodynamic averages remain unbiased; as already pioneered in diverse areas of physics (Noé et al., 2019; Wu et al., 2019; Albergo et al., 2019). This distinction motivates different algorithmic and evaluation choices

In this work, we focus on sampling amorphous particle systems — disordered arrangements of interacting particles, possibly of different subtypes, that lack the atomic periodicity of crystals. The most prominent examples are structural glasses and glass-forming liquids (Berthier & Biroli, 2011). A major obstacle to their theoretical understanding lies in the extremely long equilibration timescales, even at moderate system sizes, which severely limit the applicability of traditional numerical methods, such as molecular dynamics or local Monte Carlo, to sample from the equilibrium Boltzmann distribution (Berthier & Reichman, 2023). The state-of-the-art swap Monte Carlo algorithm extends this reach, but remains limited to specific models (Ninarello et al., 2017). These limitations highlight the potential of generative models as an alternative route to equilibrium sampling in amorphous materials.

To take into account symmetries of amorphous particle systems, prior work considered equivariant continuous normalizing flows (CNFs) trained via maximum likelihood (Jung et al., 2024), but this approach is computationally expensive. Diffusion-based models, by contrast, offer a simulation-free and scalable alternative (Yang & Schwalbe-Koda, 2025); however, while they generate visually

realistic amorphous samples, they lack tractable likelihoods and therefore cannot be used to correctly sample the Boltzmann distribution. Motivated by these limitations, we focus on flow matching and, more generally, stochastic interpolants (Albergo & Vanden-Eijnden, 2023; Albergo et al., 2023; Lipman et al., 2023), that can (i) be trained using a simulation-free mean-squared loss while they admit a CNF formulation and therefore can (ii) easily incorporate symmetries using an equivariant velocity field (Köhler et al., 2020; Garcia Satorras et al., 2021) and (iii) have a tractable likelihood to unbias the generated samples (Chen et al., 2019). Moreover, recent extensions to Riemannian manifolds (Chen & Lipman, 2024; Wu et al., 2025) make this framework particularly well suited to simulations using periodic boundary conditions (PBC), as required to minimize boundary effects in amorphous materials, a direction also explored for crystalline systems in Miller et al. (2024).

Guided by these goals, we make the following contributions:

- We introduce the equivariant Riemannian stochastic interpolant (eRSI) framework, which extends Riemannian stochastic interpolants by incorporating invariance constraints tailored to amorphous particle systems;

- We prove that the optimal marginal paths and velocity fields induced by our objective respect the full symmetry group relevant to amorphous systems with multiple species on the torus. These theoretical guarantees are original contributions to the flow matching and stochastic interpolant frameworks.

- We adapt the graph neural network of Satorras et al. (2021) to respect the full set of symmetries relevant for amorphous materials;

- We apply our framework to a canonical model of metallic glass formers widely used in theoretical and computational studies of amorphous materials, and benchmark it against symmetry- and geometry-agnostic baselines. Our approach yields higher-quality generations, both in individual snapshots and in averaged physical observables (via importance sampling reweighting), while also exhibiting improved scalability with system size.

## 2 PRELIMINARIES

### 2.1 AMORPHOUS PARTICLE SYSTEMS

**Structure of the configuration space.** A configuration of $N$ particles is denoted by $C \in \mathcal{C}$ and decomposed as $C = (s, X)$, where $s = (s_1, \ldots, s_N) \in \mathcal{S}^N$ encodes the species of each particle within a finite set $\mathcal{S}$, and $X$ specifies their spatial coordinates over $\mathcal{M}^N$ where $\mathcal{M} = [0, L)^d$ is the $d$-dimensional flat torus of length $L > 0$. On the torus, particle coordinates are not unique: for any $(s, X) \in \mathcal{C}$, all configurations of the form $(s', X')$ with $s' = s$, $X' = X + kL$, $k \in \mathbb{Z}^{Nd}$, represent the same physical state. This ambiguity is lifted by applying component-wise the modulo operator

$$A \% L = A - \left\lfloor \frac{A}{L} \right\rfloor L , \qquad A \in \mathbb{R} ,$$

mapping all equivalent coordinates back into the fundamental domain of the torus. The flat torus implements periodic boundary conditions (PBC), as particles exiting one side of the domain re-enter from the opposite side. We consider the nearest image distance $\mathrm{d}_{\mathcal{M}}$

$$\mathrm{d}_{\mathcal{M}}(X, Y) = \min_{k \in \mathbb{Z}^d} \mathrm{d}_E \left( X, Y + kL \right) , \tag{1}$$

which is a distance on $\mathcal{M}$ but not on $\mathbb{R}^d$ since it doesn't satisfy the triangular equality.

**Equilibrium distribution.** Denoting $X_{(i)} \in \mathcal{M}$ the coordinates of the $i$-th particle, the system is governed by the potential energy

$$\mathrm{U}_\star(s, X) = \sum_{i=1}^{N} \sum_{j<i}^{N} \mathrm{W} \left( s_i, s_j, \mathrm{d}_{\mathcal{M}}(X_{(i)}, X_{(j)}) \right) , \tag{2}$$

where $\mathrm{W} : \mathcal{S} \times \mathcal{S} \times \mathbb{R}^+ \to \mathbb{R}$ is a pairwise interaction potential. At temperature $T$, the equilibrium distribution is the Boltzmann measure

$$p_\star(\mathrm{d}s, \mathrm{dvol}_X) = \frac{1}{\mathcal{Z}} \exp \left( -\frac{\mathrm{U}_\star(s, X)}{k_{\mathrm{B}} T} \right) \mathrm{d}s \, \mathrm{dvol}_X , \tag{3}$$

Figure 1: Illustration of invariance group actions on a configuration of the 2D 10-particle IPL model. The system contains two particle species with different effective diameters (see Section 5). The symmetrized transformation shown corresponds to a $90°$ counterclockwise rotation – equivalently, an axial symmetry with respect to the diagonal from the bottom-left to the top-right corner.

where $k_{\mathrm{B}}$ is the Boltzmann constant, $\mathrm{dvol}_X$ is the volume element over $\mathcal{M}$ and the partition function $\mathcal{Z} = \int \exp\left(-\mathrm{U}_\star(s, X)/k_{\mathrm{B}}T\right) \mathrm{d}s \mathrm{dvol}_X$ ensures the normalization. This joint distribution implements a uniform distribution on the specie of particles and correspond to sampling from a "semi-grand" canonical ensemble in the language of statistical mechanics, where the total number of particles is fixed but the composition is not. We will also be interested in sampling from $p_\star$ conditioned on a composition. Our objective is to build a generative model that approximates $p_\star$ (or its conditional) to allow efficient sampling of equilibrium configurations.

In the following, we examine the symmetries on the multidimensional torus induced by the potential, and introduce Riemannian stochastic interpolants, the generative modeling framework underlying our approach.

## 2.2 INVARIANCES ON THE TORUS SPACE

**Definition 1.** *Let $G$ denote a set of group actions acting on the configuration space $\mathcal{C}$. A probability density $q$ on $\mathcal{C}$ is said to be $G$-invariant if, for all $g \in G$ and all $C \in \mathcal{C}$, $q(g(C)) = q(C)$ .*

The potential (2) satisfies several invariance properties that correspond to group actions on the configuration space $\mathcal{C}$, capturing symmetries of the system. These invariances are under the group actions of (see also Figure 1 illustration):

- **Permutations**, that permutes the particles and their associated species. For any permutation $\sigma \in S_N$, define

$$g_\sigma : (s, X) \mapsto \left((s_{\sigma(1)}, \ldots, s_{\sigma(N)}), (X_{(\sigma(1))}, \ldots, X_{(\sigma(N))})\right) \ .$$

- **Translations**, that translates all coordinates by the same vector and wraps them back onto $\mathcal{M}$. Denoting by $\mathbf{1}_N \in \mathbb{R}^N$ the vector with all coordinates equal to 1. For any $u \in \mathbb{R}^d$, define

$$g_u : (s, X) \mapsto (s, (X + \mathbf{1}_N \otimes u) \ \% \ L) \ ,$$

- **Symmetries**, that combines permutations of axes and sign flips of coordinates along these axes. For any signed permutation matrix $M$ in the $d$-dimensional hyperoctahedral group $B_d$, define

$$g_M : (s, X) \mapsto (s, ((\mathrm{I}_N \otimes M)X) \ \% \ L) \ ,$$

where $\mathrm{I}_N$ is the identity matrix of size $N \times N$. While particle systems such as single molecules, like proteins, may exhibit full rotational invariance, restricting coordinates to a flat torus reduces these symmetries to signed permutations of the coordinate axes.

We denote by $G_\mathcal{C}$ the group generated by the above actions and prove in Appendix A that $p_\star$ defined in Equation (3) is $G_\mathcal{C}$-invariant. Note that for any $g \in G_\mathcal{C}$, $g$ can be written as $g = f_L \circ h_{A,b}$ where

$$f_L : (s, X) \mapsto (s, X \ \% \ L) \quad \text{and} \quad h_{A,b} : C \mapsto AC + b \ , \tag{4}$$

for $A$ an orthogonal matrix and $b$ a vector in $\mathbb{R}^{N|\mathcal{S}|+Nd}$ which are defined in Lemma 10.

## 2.3 RIEMANNIAN STOCHASTIC INTERPOLANTS

**Stochastic interpolants.** Stochastic interpolants (SI) (Albergo & Vanden-Eijnden, 2023; Albergo et al., 2023) are a generative modeling framework closely related to flow matching (FM) (Liu et al., 2023; Lipman et al., 2023). SI relies on an interpolation process $(X_t)_{t \in [0,1]}$ on $\mathbb{R}^d$ between a simple base distribution ($X_0 \sim p_{\text{base}}$) and a target distribution ($X_1 \sim p_\star$). This process is defined through an interpolation function $X_t = I(t, X_0, X_1)$ satisfying the boundary conditions $I(0, X_0, X_1) = X_0$ and $I(1, X_0, X_1) = X_1$. SI then seek to estimate a time-dependent velocity field $\hat{v}$ such that the process $(\hat{X})_{t \in [0,1]}$ defined as the integration of the ODE

$$\mathrm{d}\hat{X}_t = \hat{v}(t, \hat{X}_t)\mathrm{d}t, \quad \hat{X}_0 \sim p_{\text{base}} , \tag{5}$$

shares the same time-marginal distributions as $(X_t)_{t \in [0,1]}$. An exact solution $v_t^\star$ is given by the conditional expectation

$$v^\star(t, x) = \mathbb{E}\left[\partial_t I(t, X_0, X_1) \mid X_t = x\right] , \tag{6}$$

which can also be expressed as a minimizer of a mean-squared regression loss

$$v^\star \in \arg\min_{\hat{v}} \mathcal{L}(\hat{v}), \quad \text{with} \quad \mathcal{L}(\hat{v}) = \int_0^1 \mathbb{E}\left[\|\hat{v}(t, X_t) - \partial_t I(t, X_0, X_1)\|^2\right]\mathrm{d}t . \tag{7}$$

This optimization problem allows to build an empirical loss from samples of $p_{\text{base}}$ and $p_\star$ to train a parametrized velocity field $\hat{v}$ to approximate $v^\star$. The ODE push-forward formulation of Equation (5) yields a tractable likelihood by integration of the instantaneous change-of-variables formula (Chen et al., 2019, Theorem 1). Specifically, the logarithm of the density $\hat{q}_t$ of $\hat{X}_t$ evolves along ODE solutions according to

$$\frac{\mathrm{d}}{\mathrm{d}t} \log \hat{q}_t(\hat{X}_t) = -\operatorname{div} \hat{v}(t, \hat{X}_t), \quad \log \hat{q}_0(\hat{X}_0) = \log p_{\text{base}}(\hat{X}_0) . \tag{8}$$

While a tractable likelihood enables its direct use as a training objective, it is substantially more expensive than the SI loss in Equation (7) for two reasons. First, computing the divergence requires additional auto-differentiation, although restricting to custom architectures for which the divergence is cheaper helps (Köhler et al., 2019). Second, its a not a simulation free objective which requires either the adjoint method Chen et al. (2019) or to discretize before optimization Gholaminejad et al. (2019).

**Riemannian extensions.** Recently, Chen & Lipman (2024) extended FM to manifold-supported distributions, resulting in Riemannian flow matching (RFM) shortly followed by Wu et al. (2025), who introduced Riemannian stochastic interpolants (RSI) as a manifold generalization of SI. A considered example in these papers, also leveraged by Miller et al. (2024) for crystals, is the flat torus. Following Wu et al. (2025), an interpolation strategy is to follow geodesics defined through the exponential and logarithmic maps. For the torus, they are defined as follows.

**Definition 2.** *The exponential and logarithmic maps on the d-dimensional flat torus $\mathcal{M}$ are*

$$\exp_A(V) = (A + V) \% L, \quad \log_A(B) = \left(B - A + \frac{L}{2}\right) \% L - \frac{L}{2} ,$$

*for $A, B \in \mathcal{M}$ and $V \in \mathcal{T}_A\mathcal{M}$, where $\mathcal{T}_A\mathcal{M}$ is the tangent space of the torus at point $A$ equipped with the regular Euclidian dot product. All functions and operations are applied component-wise.*

The geodesic interpolation path between $x_0$ and $x_1$ on $\mathcal{M}$ is $I_L(t, x_0, x_1) = \exp_{x_0}\left(t \log_{x_0}(x_1)\right)$. Then, given a base distribution $p_{\text{base}}$ and target distribution $p_\star$ on $\mathcal{M}$, with $X_0 \sim p_{\text{base}}$, $X_1 \sim p_\star$ and $X_t = I_L(t, X_0, X_1)$, the minimizer $\hat{v}$ of

$$\mathcal{L}^{\mathcal{M}}(\hat{v}) = \int_0^1 \mathbb{E}\left[\left\|\hat{v}(t, X_t) - \log_{X_0}(X_1)\right\|^2\right]\mathrm{d}t , \tag{9}$$

generates, by integration of the ODE in Equation (5), a stochastic process $(\hat{X})_{t \in [0,1]}$ with the same time-marginal as $(X_t)_{t \in [0,1]}$. When considering the product space $\mathcal{C}$, the RSI framework decomposes across components, i.e., it can be applied independently on species and coordinates as done by Miller et al. (2024) (see Appendix A.3).

# 3 RIEMANNIAN STOCHASTIC INTERPOLANTS FOR AMORPHOUS MATERIALS

We adapt RSI to ensure that the generative model rigorously respects the invariances of amorphous materials described in Section 2.2. First, we define the interpolation process so that it respects at all times the target's invariances. Second, we adapt an equivariant graph neural network (GNN) architecture for the torus to guarantee that the learned generative process also respects the target's invariances.

## 3.1 EQUIVARIANT RIEMANNIAN STOCHASTIC INTERPOLANTS

Considering the group $G_{\mathcal{C}}$ of symmetries of the Boltzmann distribution $p_\star$, we define equivariant interpolation functions and show that they allow building invariant interpolation processes.

**Definition 3.** *An interpolation function $I : [0,1] \times \mathcal{C} \times \mathcal{C} \to \mathcal{C}$ is said to be $G_{\mathcal{C}}$- equivariant if for all $g \in G_{\mathcal{C}}$, $I(t, g(C_0), g(C_1)) = g(I(t, C_0, C_1))$ holds for any $t \in [0,1]$ and $C_0, C_1 \in \mathcal{C}$.*

**Proposition 4.** *Given $p_{\text{base}}$ and $p_\star$ both $G_{\mathcal{C}}$-invariant distributions on $\mathcal{C}$ and a $G_{\mathcal{C}}$-equivariant interpolant, the interpolation process defined for any $t \in [0,1]$ as $X_t = I(t, X_0, X_1)$ with $X_0 \sim p_{base}$ and $X_1 \sim p_\star$ has $G_{\mathcal{C}}$-invariant time-marginal densities.*

Proofs are provided in Appendix B. In cases where $\mathcal{S}$ is a convex set, we show in Proposition 28 from the same appendix that a simple example of a $G_{\mathcal{C}}$-equivariant interpolant is the geodesic interpolant on the product space

$$I(t, (s_0, X_0), (s_1, X_1)) = \begin{pmatrix} (1-t)s_0 + ts_1 \\ \exp_{X_0}\left(t \log_{X_0}(X_1)\right) \end{pmatrix} . \tag{10}$$

## 3.2 EQUIVARIANT VELOCITY FIELD ON MULTI-COMPONENT PARTICLE SYSTEMS

We now consider a velocity field $\hat{v}$ on $\mathcal{C}$ and denote by $\hat{T}_t$ the map transporting an initial configuration of particles along the velocity field between time $0$ and time $t$. In order words, $\hat{T}_t(c_0)$ gives the solution of the ODE $dC_u = \hat{v}(u, C_u)du$ at time $t$ for the initial condition $C_0 = c_0$. Given a base distribution $p_{\text{base}}$, we seek to learn $\hat{v}$ such that the push-forward of $p_{\text{base}}$ through the transport map $\hat{T}_t$, which is the marginal distribution of the RSI model denoted by $\hat{q}_t$ below, is $G_{\mathcal{C}}$-invariant.

For this, we rely on Proposition 6 extending the result of (Köhler et al., 2020, Theorem 1) to the invariance group $G_{\mathcal{C}}$ and proven in Appendix C.

**Definition 5.** *A diffeomorphism $T : \mathcal{C} \to \mathcal{C}$ is $G_{\mathcal{C}}$-equivariant if for all $g \in G_{\mathcal{C}}$, $T \circ g = g \circ T$ holds.*

**Proposition 6.** *Let $q$ be a density of probability on $\mathcal{C}$ and let $T$ be a diffeomorphism on $\mathcal{C}$. If $q$ is $G_{\mathcal{C}}$-invariant and $T$ is $G_{\mathcal{C}}$-equivariant, the push-forward of $q$ through $T$ is also $G_{\mathcal{C}}$-invariant.*

Assuming the set of particle species $\mathcal{S}$ is bounded, a simple choice of $G_{\mathcal{C}}$-invariant $p_{\text{base}}$ is the uniform distribution on $\mathcal{S} \times \mathcal{M}^N$. To make sure that the $\hat{T}_t$ are $G_{\mathcal{C}}$-equivariant, Proposition 8 is adapted from (Köhler et al., 2020, Theorem 2).

**Definition 7.** *A velocity field $\hat{v} : [0,1] \times \mathcal{C} \to \mathcal{TC}$ is $G_{\mathcal{C}}$-equivariant if, for all $g \in G_{\mathcal{C}}$, using the decomposition $g = f_L \circ h_{A,b}$ (with $A$ and $b$ defined in Lemma 10),*

$$\hat{v}(t, g(C)) = A\,\hat{v}(t, C), \quad \forall C \in \mathcal{C} .$$

**Proposition 8.** *If $\hat{v} : [0,1] \times \mathcal{C} \to \mathcal{TC}$ is a Lipschitz-bounded $G_{\mathcal{C}}$-equivariant velocity field, then the transport maps $\hat{T}_t$ induced by its ODE flow are $G_{\mathcal{C}}$-equivariant for all times $t \in [0,1]$.*

**Proposition 9.** *Given a $G_{\mathcal{C}}$-equivariant interpolation function $I$, if $p_{base}$ and $p_\star$ are $G_{\mathcal{C}}$-invariant, then the corresponding $v^\star$ (see Equation (6)) is $G_{\mathcal{C}}$-equivariant.*

Note that Proposition 9, proven in Appendix D, together with the invariance of $p_{\text{base}}$, directly implies the invariance of the marginal paths described in Proposition 4.

Proposition 4 and Proposition 9 are original theoretical contributions to the stochastic interpolant and flow matching frameworks. These results establish the precise conditions under which group equivariance is preserved by both the interpolation and the corresponding optimal velocity field. Prior

work has largely relied on equivariant flows Köhler et al. (2019) extended to incorporate equivariances into the optimal transport alignment Klein et al. (2023). Our work instead show that equivariance arises intrinsically from the structure of the interpolant, providing a principled foundation for the architectures developed below.

In order to model $v^\star$, which is $G_\mathcal{C}$-equivariant, we follow recent work on equivariant architectures (Köhler et al., 2020; Satorras et al., 2021; Jiao et al., 2023; Miller et al., 2024), we parameterize the velocity field $\hat{v}$ using a family of $G_\mathcal{C}$-equivariant GNNs. The construction adapts the architecture by Satorras et al. (2021) to the torus geometry. Position variables are initialized in the input configuration $X_{(i)}^0 = X_{(i)}$ and particle features embed time and particle specie $H_i^0 = (t, s_i)$. The GNN architecture then iterates from layer $k$ to layer $k+1$

$$M_{ij}^k = \hat{\phi}_e(H_i^k, H_j^k, \mathrm{d}_\mathcal{M}(X_{(i)}, X_{(j)})^2) \, , \tag{11}$$

$$P_i^k = \sum_{i \neq j} \hat{\phi}_m(M_{ij}^k) M_{ij}^k, \quad H_i^{k+1} = \hat{\phi}_h(H_i^k, P_i^k) \, , \tag{12}$$

$$X_{(i)}^{k+1} = \exp_{X_{(i)}^k} \left( \sum_{i \neq j} \frac{\log_{X_{(j)}^k} X_{(i)}^k}{\mathrm{d}_\mathcal{M}(X_{(i)}, X_{(j)}) + 1} \hat{\phi}_d(M_{ij}^k) \right) \, , \tag{13}$$

where the $\hat{\phi}_.$ are neural networks such that $\hat{\phi}_e$ outputs edges features in $\mathbb{R}^n$ representing pairwise interactions between particles, $\hat{\phi}_m$ transforms edge messages while preserving dimension before aggregation, $\hat{\phi}_h$ updates the particle features and $\hat{\phi}_d : \mathbb{R}^n \to \mathbb{R}^d$ decodes interaction features into displacements on the torus. For a GNN of depth $K$, the velocity field is finally obtained as

$$\hat{v}(t, C) = \begin{pmatrix} \mathbf{0}_{N \times d_s} \\ \log_{X_{(1)}} X_{(1)}^K, \ldots, \log_{X_{(N)}} X_{(N)}^K \end{pmatrix} \tag{14}$$

We show in Proposition 34 of Appendix D that this architecture defines a Lipschitz-bounded $G_\mathcal{C}$-equivariant velocity field on $\mathcal{C}$. In Appendix B, Figure 5 compares two ODE trajectories obtained with such a velocity field and that were started on two base configurations that differ by the application of $G_\mathcal{C}$ actions.

### 3.3 SAMPLING CONDITIONED ON A FIXED COMPOSITION OF SPECIES.

**Fixing composition.** To study the properties of materials with a given composition, the number of particle per species must be fixed. The resulting conditional of $p_\star$ corresponds to the canonical distribution which remains $G_\mathcal{C}$-invariant (a proof of this invariance is provided in Proposition 22 of Appendix A). For this task, the base distribution is defined as the product of a uniform distribution over permutations of species with the target composition and a uniform distribution on $\mathcal{M}^N$. During training, a permutation can always be found such that $s_0 = s_1$. The $s$-component of the interpolation process $C_t$ (see Equation (10)) is kept constant over time. Accordingly, the architecture in Equation (14) enforces a zero velocity on the $s$-component.

**Equivariant optimal transport.** Recent works improved stochastic interpolants by seeking to replace the independent endpoint sampling $(X_0, X_1) \sim p_{\text{base}} \otimes p_\star$ with a coupling closer to the optimal transport (OT) coupling $(X_0, X_1) \sim \Pi(p_{\text{base}}, p_\star)$ (Tong et al., 2024; Albergo et al., 2024). To ease computations, the OT problem is solved between mini-batches of samples from $p_{\text{base}}$ and $p_\star$ used when computing the training objective (9), as described in (Fatras et al., 2021), which is an idea widely adopted for particle systems (Klein et al., 2023; Song et al., 2023; Irwin et al., 2024). Given two configurations $C_0 = (\tilde{s}, X_0)$ and $C_1 = (\tilde{s}, X_1)$ with identical composition, each is partitioned by species, and within each group the Hungarian algorithm is applied using $\mathrm{d}_\mathcal{M}$ (see Equation (1)) as cost. This OT-based matching, consistent with the particle-permutation invariance, shortens particle transport paths significantly. Note that Klein et al. (2023) accounts for other invariances in their solution of the OT problem. In the experiments below we only considered particle-permutation invariance to ease the computational budget.

## 4 RELATED WORKS

**Generative modeling for equilibrium Boltzmann sampling.** Sampling equilibrium configurations is substantially more difficult than tasks that have been more widely studied by the generative modeling community, such as predicting single molecular structures (e.g., proteins or drug–ligand complexes) or predicting crystalline structures. Instead of predicting one plausible configuration – typically an energy or free energy minimum, one needs to accurately reproduce the correct statistics, which includes entropic effects and, possibly, multiple modes. Since a crucial requirement in this context is tractable likelihood estimation to enable the computation of unbiased physical observables, early advances leveraged normalizing flows (Noé et al., 2019; Albergo et al., 2019) and autoregressive models (Wu et al., 2019) (citing pioneering works followed by many more, see Coretti et al. (2024) for a partial review). Here, we focus on ODE-based generative models, which offer greater expressivity and more flexibility for incorporating invariances.

**Invariances in generative modeling.** Incorporating invariances into push-forward generative models has been studied in Köhler et al. (2019); Köhler et al. (2020); Wirnsberger et al. (2022); Midgley et al. (2023); Biloš & Günnemann (2021), with Klein et al. (2023) adapting FM to build equivariant transport maps, deriving results akin to Proposition 8, but limited to linear actions on $\mathbb{R}^n$. In parallel, Denoising Diffusion Probabilistic Models (DDPMs) were extended with equivariances (Xu et al., 2022; Hoogeboom et al., 2022), enabling generation but not likelihood evaluation. All these advances relied on equivariant GNNs (Satorras et al., 2021; Garcia Satorras et al., 2021), which also underpin our approach. Interestingly, later works showed that strong generative performance could be obtained even without explicit invariance constraints (Martinkus et al., 2023; Chu et al., 2024; Joshi et al., 2025), suggesting that symmetries enhance efficiency and generalization, though they are not strictly required for generation. Our experiments below demonstrate that incorporating invariances is advantageous.

**Generative models for crystalline and amorphous materials.** Generative modeling for materials has attracted significant attention, in particular for crystals. Like the systems considered here, crystals exhibit a number of symmetries and non-Euclidean representations. Most approaches build on Riemannian extensions of DDPMs (Yang et al., 2023; Jiao et al., 2023; 2024; Zeni et al., 2025; Levy et al., 2025), while works such as Miller et al. (2024); Sriram et al. (2024) resonate more closely with our setting by embedding invariances into RFM. However, these methods focus on generative modeling from samples drawn from an unknown distribution. In this context, success is typically assessed by visual or structural fidelity to the training configurations, which makes the task substantially easier than sampling from a known energy, where likelihoods or importance weights are required for unbiased observable estimation. Moreover, unlike amorphous materials, crystals can be described by unit cells and fractional coordinates, due to their atomic periodicity, which entail distinct invariance structures.

For amorphous systems, Li et al. (2025) proposed a Riemannian DDPM approach (unsuitable for unbiased observable computation due to its lack of tractable likelihood), while Jung et al. (2024) performs maximum-likelihood training of ODE flows which is computationally more expensive than objective (9) due to the divergence term in the model's likelihood (8). Köhler et al. (2020) alleviates this issue by restricting the velocity field to a family of equivariant parameterizations with cheap divergence computations, but this reduces expressivity.

## 5 NUMERICAL EXPERIMENTS

We test our framework on an amorphous particle system designed to remain non-crystalline and to form glasses at low temperatures. This system is a paradigmatic model of metallic glass formers, extensively used in theoretical and computational studies of the glass transition. Even at modest system sizes, it exhibits the characteristic hallmarks of complex glassy behavior—slow relaxation and heterogeneous dynamics—while remaining challenging to sample with standard simulation techniques at low temperatures. Comparing to alternative frameworks and architectures, we demonstrate the benefit of enforcing periodic boundary conditions and symmetries. Looking forward, we also investigate the impact of increasing the system size.

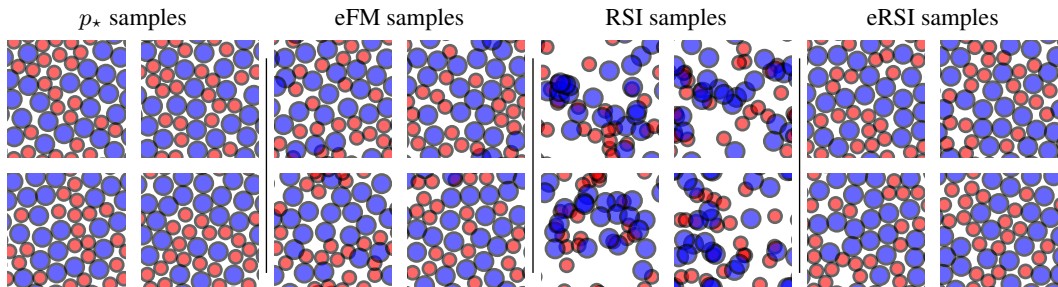

Figure 2: **Samples from $p_\star$ and compared generative models on the 44-particle IPL system.** eFM denotes a model trained with standard FM that uses an equivariant velocity respecting system symmetries but ignores the torus geometry, thus generating unphysical particle overlaps near boundaries. RSI uses a non-equivariant velocity field. eRSI is the proposed approach, combining RSI with an equivariant velocity field.

**System.** We consider a two-dimensional binary mixture with fixed composition of equal fractions of two species with pairwise interactions obeying inverse power laws (IPL), a classical glass model (Bernu et al., 1987; Perera & Harrowell, 1999). For two particles at nearest image distance $r$ with respective species $s_1$ and $s_2$, the interaction potential is

$$W_{\mathrm{IPL}}(s_1, s_2, r) = \begin{cases} \epsilon \left( \dfrac{\sigma_{s_1 s_2}}{r} \right)^{12} + W_0, & r < 2.5\sigma, \\ 0, & \text{otherwise,} \end{cases}, \quad \text{with } \sigma = \begin{pmatrix} 1.0 & 1.2 \\ 1.2 & 1.4 \end{pmatrix}, \ \epsilon = 1 \ .$$

The value $W_0$ is chosen to shift the potential continuously to zero at $r^{\mathrm{cut}} = 2.5\sigma$. We consider two system sizes $N \in \{10, 44\}$ and define the box length $L$ such that the number density $N/L^2$ is fixed to 0.5. The temperature is $T = 0.1$. Samples for the $N = 44$ system are shown in Figure 2.

**Framework comparison.** The equivariant RSI (eRSI) framework introduced in Sections 2.3 and 3, is compared with two following ablations: (i) equivariant FM (eFM) (Klein et al., 2023), which applies standard flow matching with an equivariant GNN (Satorras et al., 2021) respecting permutation–translation–rotation symmetries in $\mathbb{R}^{Nd}$ but ignoring torus geometry; and (ii) RSI, which accounts for torus geometry but lacks the invariances of Section 3 using a simple multi-layer perceptron to learn the velocity field. For a a fair comparison, the number of trainable parameters of all architectures is taken equal and all models are trained for the same number of epochs. Training datasets of $10^5$ independent configurations from $p_\star$ with the desired equal fractions of each species are generated using long Markov Chain Monte Carlo simulations (see Appendix F.1 for details). We refer to Appendix F for implementation details.

**Quantities of interests.** The different models are compared through physical observables of interest for amorphous systems. We measure (i) the average potential energy $U = \mathbb{E}_{p_\star}[U_\star]$ of generated configurations, (ii) the specific heat $c_V = (\mathbb{E}_{p_\star}[U_\star^2] - \mathbb{E}_{p_\star}[U_\star]^2)/(Nk_{\mathbf{B}}T^2)$, probing energy fluctuations and known to be harder to recover than average energies (Flenner & Szamel, 2006; Jung et al., 2024), and (iii) the radial distribution function (Barrat & Hansen, 2003) which measures the density profile around a tagged particle

$$g(r) = \mathbb{E}_{p_\star} \left[ \frac{L^2}{N^2} \sum_{i=1}^{N} \sum_{\substack{j=1 \\ j \neq i}}^{N} \delta\left( r - \mathrm{d}_{\mathcal{M}}(X_{(i)}, X_{(j)}) \right) \right] \ .$$

To compare the different models, we evaluate the averages of physical observables using importance sampling (IS) estimation and present snapshots of the generated configurations. As long as the target distribution's support is contained within that of the model, IS enables reweighting of model samples to yield asymptotically unbiased estimators (see Appendix E for details). Reference values (reported as Target) are computed using ground-truth samples generated by gold-standard samplers, as described in Appendix F.1. In Figures 3 and 4, we assess the sampling efficiency of IS reweighting by varying the number of model samples $R$ used in the estimators.

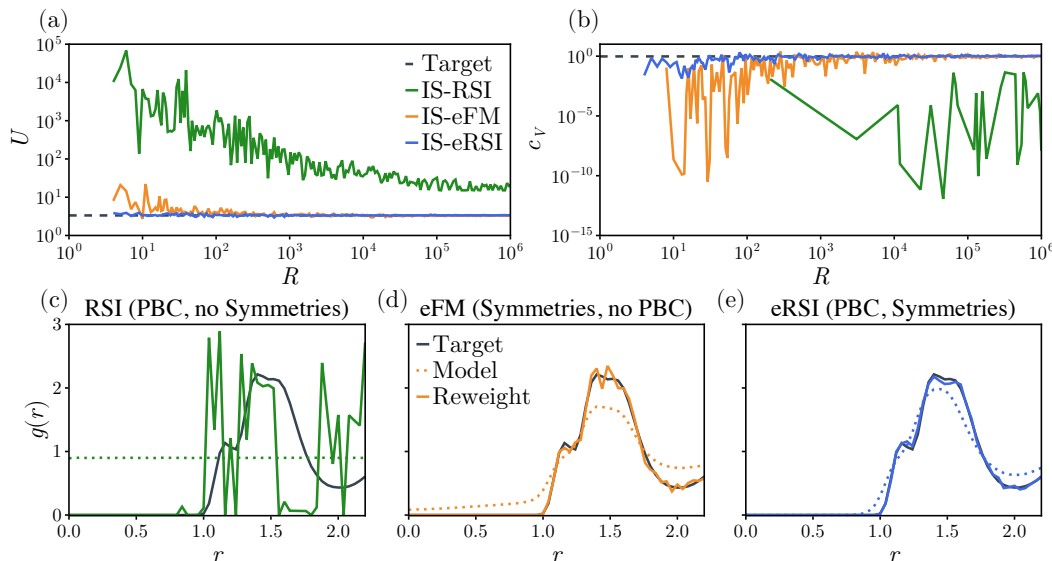

Figure 3: **Results for $N = 10$ particles.** (a) Mean potential energy $U$ and (b) specific heat $c_V$ as a function of the number of generated samples $R$ for RSI, eFM, and eRSI. (c–e) Radial distribution function $g(r)$ for the three models, showing target, direct model, and reweighted estimates. RSI fails completely, eFM partially recovers observables, and eRSI performs best.

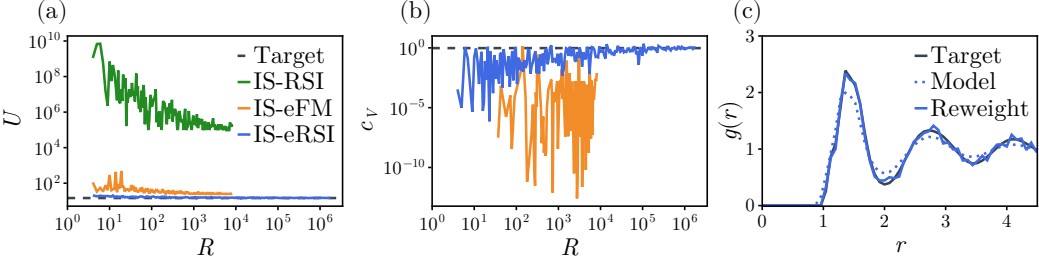

Figure 4: **Results for $N = 44$ particles.** (a) Mean potential energy $U$, and (b) specific heat $c_V$ as a function of the number of generated samples $R$ for RSI, eFM, and eRSI. (c) Radial distribution function $g(r)$ for eRSI averaged over $1.8 \times 10^6$ samples. RSI and eFM deviate due to collapsed states (RSI) or boundary overlaps (eFM). RSI samples were of too poor quality to produce $c_V$ estimates. Only eRSI remains consistent with the target distribution.

**Results: averages of physical observables**   Figure 3 presents results for the small system with $N = 10$ particles. RSI fails to capture meaningful structure, as indicated by the flat $g(r)$. In contrast, eFM performs better: $U$ and $c_V$ can be recovered through reweighting, along with a noisy estimate of $g(r)$. eRSI achieves even stronger performance: $U$ and $c_V$ converge to the target with fewer samples, and the estimate of $g(r)$ is more accurate. We repeat the analysis for $N = 44$, with results shown in Figures 2 and 4. Here, eFM fails: both $c_V$ and $U$ estimators stabilize away from the ground truth (sampling was stopped at $R = 8 \times 10^3$ due to limited computational resources). In contrast, eRSI produces $U$ and $c_V$ that converge rapidly to the target, together with an accurate $g(r)$. For $g(r)$, we also include the poor estimates obtained without reweighting (dotted lines), highlighting both the importance of reweighting and the need for models that can handle it accurately.

**Results: snapshots of generated configurations**   Examples of configurations generated by the different models for $N = 44$ are shown in Figure 2, alongside typical configurations from $p_\star$. RSI fails to move particles, producing configurations that remain close to the uniform base distribution—consistent with its poor performance. Augmenting the training data with random actions from

$G_{\mathcal{C}}$ to help the RSI model learn the invariances was attempted, but this did not improve performance. eFM generates more realistic configurations, but many particle overlaps occur near the box edges because periodic boundary conditions are ignored. These overlaps cause three issues: (i) averages obtained without reweigthing display unphysical characteristics, as a much higher value of $g(r < 1)$, (ii) many configurations are discarded during reweighting, as configurations with close particle pairs receive zero weight in the IS estimator (thus requiring a large number of samples to obtain accurate estimation), and (iii) the likelihood of overlaps at the boundaries increases with system size, limiting the scalability of the eFM model. In contrast, configurations produced by eRSI correctly incorporate periodic boundary conditions and do not suffer from this problem.

## 6 CONCLUSION AND LIMITATIONS

Our results, based on a specific model for amorphous systems, demonstrate that incorporating geometry and symmetries makes it possible to reliably estimate physical observables with fewer samples and on larger systems than comparable generative modeling frameworks that lack these properties. Although we do not anticipate major issues, these results need to be confirmed on other amorphous systems, in particular in three dimensions. The computational bottleneck of our approach for computing physical observables is the evaluation of likelihoods on large batches of particles in the IS estimation. However, the hope is that this computational cost scales more favorably than usual local Monte Carlo simulation techniques Berthier & Reichman (2023). Finally, while we used extensive training sets in our experiments, we shall next investigate training methods that alleviate this requirement, such as adaptive MCMCs Gabrié et al. (2022) possibly combined with sequential tempering Wu et al. (2019); McNaughton et al. (2020); Bono et al. (2025) as previously proposed for normalizing flows and autoregressive models.

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

# A    INVARIANCES OF THE PRE-METRIC $d_{\mathcal{M}}$ AND IMPLICATIONS

**Lemma 10.** *Let $g \in G_{\mathcal{C}}$. The action can be decomposed as*

$$g : \begin{pmatrix} s \\ X \end{pmatrix} \mapsto \begin{pmatrix} g^s(C) \\ g^X(C) \end{pmatrix} = \begin{pmatrix} A_\sigma s \\ (A_\sigma \otimes I_d) h_{M,c,L}(X) \end{pmatrix} ,$$

$$h_{M,c,L}(X) = ((I_N \otimes M)X + (\mathbf{1}_N \otimes c)) \% L ,$$

*where $\sigma \in S_N$ is a permutation, $A_\sigma$ is the associated permutation matrix of size $N \times N$, $M$ is a signed permutation matrix of size $d \times d$ and $c$ is a vector in $\mathbb{R}^d$. The function $h_{M,c,L}$ could be also un-vectorized as*

$$h_{M,c,L}(X) = \left\{ \tilde{h}_{M,c,L}(X_{(i)}) \right\}_{i=1}^N , \quad \tilde{h}_{M,c,L}(y) = (My + c) \% L ,$$

*which also means that for all $i \in [\![1, N]\!]$*

$$g^X(C)_{(i)} = \tilde{h}_{M,c,L}(X_{(\sigma(i))}) .$$

The set of actions induced by $h_{M,c,L}$ on $\mathcal{M}^N$ is denoted $\mathcal{G}_{\mathcal{M}^N}$ and the one induced by $\tilde{h}_{M,c,L}$ on $\mathcal{M}$ is denoted $\mathcal{G}_{\mathcal{M}}$. From Lemma 10, we also deduce that $A$ and $b$ in Equation (4) can be written as

$$A = \begin{pmatrix} A_\sigma & 0 \\ 0 & (A_\sigma \otimes I_d)(I_N \otimes M) \end{pmatrix} , \quad b = \mathbf{1}_N \otimes c . \tag{15}$$

**Lemma 11.** *Let $X, Y \in \mathbb{R}^{Nd}$. We have*

$$((X \% L) + Y) \% L = (X + Y) \% L . \tag{16}$$

*Proof.* We have that $X = (X \% L) + k_X L$ where $k_X \in \mathbb{Z}^{Nd}$ which leads to

$$((X \% L) + Y) \% L = (X + Y - k_X) \% L = (X + Y) \% L .$$

$\square$

**Lemma 12.** *Let $y \in \mathbb{R}$ and $\epsilon \in \{-1, +1\}$, then $(\epsilon y) \% L = (\epsilon (y \% L)) \% L$.*

*Proof.* If $\epsilon = 1$ this is obviously true. If $\epsilon = -1$, we are trying to prove that

$$(-y) \% L = (-(y \% L)) \% L .$$

By definition, we have that

$$(-y) \% L = -y - \left\lfloor \frac{-y}{L} \right\rfloor L ,$$

$$(-(y \% L)) \% L = -y - \left( \left\lfloor \frac{-y}{L} + k \right\rfloor - k \right) L ,$$

where $k = \lfloor y/L \rfloor$. Using the fact that for any $x \in \mathbb{R}$ and $n \in \mathbb{N}$,

$$\lfloor x + n \rfloor - n = \lfloor x \rfloor ,$$

we get the intended result. $\square$

**Lemma 13.** *Let $M \in \mathcal{B}_d$ and $X \in \mathbb{R}^d$, then $(MX) \% L = (M [X \% L]) \% L$.*

*Proof.* For any $M \in \mathcal{B}_d$, there exist $\epsilon \in \{-1, 1\}^d$ and $\sigma \in S_d$ such that for all $i, j \in [\![1, d]\!]$, $M_{i,j} = \epsilon(i)\delta_{\sigma(i),j}$. For all $i \in [\![1, d]\!]$, we have

$$((MX) \% L)_i = (\epsilon(i)X_{\sigma(i)}) \% L, \quad (M [X \% L])_i = \epsilon(i) [X \% L]_{\sigma(i)} .$$

Use Lemma 12 on each coordinate to conclude. $\square$

**Proposition 14.** *Let $X, Y \in \mathcal{M}$, a pre-metric (see (Chen & Lipman, 2024, Section 3.2) for the definition) between $X$ and $Y$ in $\mathcal{M}$ can be defined as*

$$d_{\mathcal{M}}(X, Y) = \min_{k \in \mathbb{Z}^d} d^E(X, Y + kL) , \tag{17}$$

*where $d^E(X, Y) = \|X - Y\|$ is the euclidean distance.*

*Proof.* Let $X, Y \in \mathcal{M}$, we obviously have that $d_{\mathcal{M}}(X, Y) \geq 0$ checking the non-negativity. Moreover, using the fact that $d^E$ is a distance

$$d_{\mathcal{M}}(X, Y) = 0 \iff \exists k \in \mathbb{Z}^d, d^E(X, Y + kL) = 0 \iff X = Y \% L \iff X = Y ,$$

which checks positivity. Using Proposition 18, we have that $\nabla d_{\mathcal{M}}(X, Y) = \log_X Y$, which leads to

$$
\begin{aligned}
\nabla d_{\mathcal{M}}(X, Y) \neq 0 &\iff \log_X Y \neq 0 \iff \left(Y - X + \frac{L}{2}\right) \% L - \frac{L}{2} \neq 0 \\
&\iff \exists! k \in \mathbb{Z}^{Nd}, \ Y - X + \frac{L}{2} + kL - \frac{L}{2} \neq 0 \\
&\iff (Y - X) \% L \neq 0 \\
&\iff X \neq Y .
\end{aligned}
$$

$\square$

**Proposition 15.** *The logarithmic map can be written for any $X \in \mathcal{M}$ and $Y \in \mathcal{T}_X \mathcal{M}$*

$$\log_X(Y) = \frac{L}{2\pi} \operatorname{atan2}\left(\sin\left(\frac{2\pi}{L}(X - Y)\right), \cos\left(\frac{2\pi}{L}(X - Y)\right)\right) .$$

*Proof.* For any $X \in \mathcal{M}$ and $Y \in \mathcal{T}_X \mathcal{M}$, we have that

$$
\begin{aligned}
\log_X(Y) &= \left(Y - X + \frac{L}{2}\right) \% L - \frac{L}{2} \\
&= Y - X - L \left\lfloor \frac{Y - X + \frac{L}{2}}{L} \right\rfloor , \\
&= \frac{L}{\pi} \arctan\left(\tan\left(\frac{\pi}{L}(Y - X)\right)\right) , &&\left(\text{Using } \arctan \tan \alpha = \alpha - \pi \left\lfloor \frac{\alpha}{\pi} + \frac{1}{2} \right\rfloor\right) \\
&= \frac{L}{\pi} \arctan\left(\frac{\sin\left(\frac{2\pi}{L}(Y - X)\right)}{1 + \cos\left(\frac{2\pi}{L}(Y - X)\right)}\right) , &&\left(\text{Using } \tan \alpha = \frac{\sin 2\alpha}{1 + \cos 2\alpha}\right) \\
&= \frac{L}{2\pi} \operatorname{atan2}\left(\sin\left(\frac{2\pi}{L}(X - Y)\right), \cos\left(\frac{2\pi}{L}(X - Y)\right)\right) .
\end{aligned}
$$

$\square$

**Lemma 16.** *For all $a \in \mathbb{R}$, $-\lfloor \frac{a}{L} + \frac{1}{2} \rfloor = \arg\min_{k \in \mathbb{Z}} |a + kL|$ holds.*

*Proof.* By definition, we have that

$$\left\lfloor \frac{a}{L} \right\rfloor L \leq a \leq \left(\left\lfloor \frac{a}{L} \right\rfloor + 1\right) L ,$$

which means that

$$k^\star = \arg\min_{k \in \mathbb{Z}} |a + kL| \in \left\{-\left\lfloor \frac{a}{L} \right\rfloor, -\left(\left\lfloor \frac{a}{L} \right\rfloor + 1\right)\right\} .$$

Case 1: $k^\star = -\lfloor a/L \rfloor$ This means that $a/L$ is closer to $\lfloor a/L \rfloor$ than to $\lfloor a/L \rfloor + 1$ which implies that $a/L + 1/2$ is also closer to $\lfloor a/L \rfloor$ which means that

$$\left\lfloor \frac{a}{L} \right\rfloor = \left\lfloor \frac{a}{L} + \frac{1}{2} \right\rfloor ,$$

and that $k^\star = -\lfloor a/L + 1/2 \rfloor$.

Case 2: $k^\star = -(\lfloor a/L \rfloor + 1)$ This means that $a/L$ is closer to $\lfloor a/L \rfloor + 1$ than to $\lfloor a/L \rfloor$ which implies that $a/L + 1/2$ is also closer to $\lfloor a/L \rfloor + 1$ which means that

$$\left\lfloor \frac{a}{L} \right\rfloor + 1 = \left\lfloor \frac{a}{L} + \frac{1}{2} \right\rfloor \; ,$$

and that $k^\star = -\lfloor a/L + 1/2 \rfloor$. □

**Lemma 17.** *Let $f : \mathbb{R}^k \to \mathbb{R}^+$ be defined for all $x \in \mathbb{R}^k$ as*

$$f(x) = \sum_{i=1}^{k} f_i(x_i), \quad f_i : \mathbb{R} \to \mathbb{R}^+ \; .$$

*Then the minimum of $f$ decomposes as*

$$\arg\min_{x \in \mathbb{R}^k} f(x) = \arg\min_{x_1 \in \mathbb{R}} f_1(x) \times \ldots \times \arg\min_{x_k \in \mathbb{R}} f_k(x) \; .$$

*Proof.* Suppose that there exist $\bar{x} \in \arg\min_{x \in \mathbb{R}^k} f(x)$ and $\bar{x} \notin \arg\min_{x_1 \in \mathbb{R}} f_1(x) \times \ldots \times \arg\min_{x_k \in \mathbb{R}} f_k(x)$. Then there exists $i \in \{1, \ldots, k\}$ such that $\bar{x}_i \notin \arg\min_{x_i \in \mathbb{R}} f_i(x)$. Let $\hat{x}_i \in \arg\min_{x_i \in \mathbb{R}} f_i(x)$, then $\hat{x} \in \mathbb{R}^k$ defined as

$$\hat{x}_j = \begin{cases} \bar{x}_j & \text{if } j \neq i \\ \hat{x}_i & \text{otherwise} \end{cases} \; .$$

Then $f(\hat{x}) \leq f(\bar{x})$ which contradicts the assumptions. This gives the left-right inclusion, and the right-left one is trivial. □

**Proposition 18.** *Let $X, Y \in \mathcal{M}$, the pre-metric $\mathrm{d}_{\mathcal{M}}$ can be written as*

$$\mathrm{d}_{\mathcal{M}}(X, Y) = \|\log_X(Y)\|, \quad \text{for any } X, Y \in \mathcal{M} \; .$$

*Proof.* We have that $\log_X(Y) = Y - X + k_{X,Y} L$ where

$$k_{X,Y} = -\left\lfloor \frac{Y - X + \frac{L}{2}}{L} \right\rfloor = -\left\lfloor \frac{Y - X}{L} + \frac{1}{2} \right\rfloor \; .$$

Moreover, we have that

$$\mathrm{d}_{\mathcal{M}}(X, Y) = \min_{k \in \mathbb{Z}^d} \|X - Y + kL\| = \min_{k \in \mathbb{Z}^d} \|Y - X + kL\| = \min_{k \in \mathbb{Z}^d} \|Y - X + kL\|^2 \; ,$$

which implies that

$$\mathrm{d}_{\mathcal{M}}(X, Y) = \min_{k \in \mathbb{Z}^d} \sum_{i=1}^{d} (Y_i - X_i + k_i L)^2 \; .$$

Using Lemma 16 and the monotonicity of the square root on $\mathbb{R}^+$, we have that that for all $i \in [\![1, d]\!]$

$$\arg\min_{k \in \mathbb{Z}} (Y_i - X_i + kL)^2 = [k_{X,Y}]_i \; .$$

By Lemma 17, this implies that

$$\arg\min_{k \in \mathbb{Z}^d} \sum_{i=1}^{d} (Y_i - X_i + k_i L)^2 = k_{X,Y} \; ,$$

leading to

$$d(X, Y) = \|Y - X + k_{X,Y} L\| = \|\log_X(Y)\| \; .$$

□

**Proposition 19.** *Let $X, Y \in \mathcal{M}$. For any $M \in B_d$,*

$$\log_{(MX) \% L}((MY) \% L) = M \log_X(Y) ,$$

*holds.*

*Proof.* Let $x, y \in [0, L]$, we have that

$$\log_{(-x) \% L}((-y) \% L) = \frac{L}{\pi} \arctan\left( \frac{\sin\left( \frac{2\pi}{L} ((-y) \% L - (-x) \% L) \right)}{1 + \cos\left( \frac{2\pi}{L} ((-y) \% L - (-x) \% L) \right)} \right).$$

By definition, there exists $k, k' \in \mathbb{Z}$ such that

$$(-x) \% L = -x + kL, \quad \text{and } (-y) \% L = -y + k'L .$$

This leads to

$$\frac{2\pi}{L} ((-y) \% L - (-x) \% L) = \frac{2\pi}{L} ([-y + k'L] - [-x + kL]) ,$$

$$= -\frac{2\pi}{L}(y - x) + 2\pi \underbrace{(k - k')}_{\in \mathbb{Z}} .$$

Using the periodicity of the sine and cosine in the above formula, we get

$$\log_{(-x) \% L}((-y) \% L) = \frac{L}{\pi} \arctan\left( \frac{-\sin\left( \frac{2\pi}{L}(y - x) \right)}{1 + \cos\left( \frac{2\pi}{L}(y - x) \right)} \right) = -\log_x(y).$$

Let $M \in B_d$ and $X, Y \in \mathcal{M}$. By definition, for all $i \in [\![1, d]\!]$, $(MX)_i = \epsilon_i X_{\sigma(i)}$. Moreover, using the previous result

$$\left[ \log_{(MX) \% L}((MY) \% L) \right]_i$$
$$= \log_{[(MX) \% L]_i}([(MY) \% L]_i) = \log_{([MX]_i) \% L}(([MY]_i) \% L) ,$$
$$= \log_{(\epsilon_i X_{\sigma(i)}) \% L}((\epsilon_i Y_{\sigma(i)}) \% L) = \epsilon_i \log_{X_{\sigma(i)}}(Y_{\sigma(i)}) ,$$
$$= [M \log_X(Y)]_i ,$$

which concludes the proof. $\qquad\square$

**Corollary 20.** *Let $X, Y \in \mathcal{M}$. For any $M \in B_d$ and $u \in \mathbb{R}^d$,*

$$\log_{(MX+u) \% L}((MY + u) \% L) = M \log_X(Y). \tag{18}$$

*Proof.* Using Lemma 11 applied twice, we have

$$\log_{(X+u) \% L}((Y + u) \% L) = \left( (Y + u) \% L - (Y + u) \% L + \frac{L}{2} \right) \% L - \frac{L}{2} ,$$

$$= \left( Y + u - Y - u + \frac{L}{2} \right) \% L - \frac{L}{2} ,$$

$$= \log_{X \% L}(Y \% L) .$$

The result comes by applying this remark in Proposition 19 with $MX \in \mathcal{M}$ and $MY \in \mathcal{M}$. $\quad\square$

**Corollary 21.** *The pre-metric $\mathrm{d}_{\mathcal{M}}$ is $\mathcal{G}_{\mathcal{M}}$-invariant, i.e, for all $X, Y \in \mathcal{M}$*

$$\mathrm{d}_{\mathcal{M}}(g(X), g(Y))) = \mathrm{d}_{\mathcal{M}}(X, Y), \quad \text{for all } g \in \mathcal{G}_{\mathcal{M}} .$$

*Consequently, the potential $\mathrm{U}_\star$ (2) and the induced density $\mathrm{p}_\star$ (3) are both $G_{\mathcal{C}}$-invariant.*

*Proof.* This comes from the fact that, according to Proposition 18, the distance can be written as the norm of the logarithmic map, which itself is equivariant as per Corollary 20 and using the decomposition of the action in Lemma 10. Using the fact that the scaling matrix is always orthogonal, we get the invariances with respect to symmetries and translations. For the permutation invariance, we just use the permutation invariance of the sum. $\qquad\square$

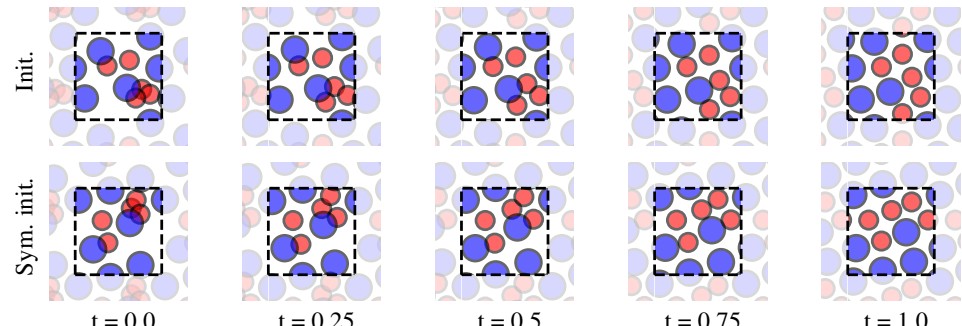

Figure 5: Two trajectories of particle configurations generated by ODE integration with an equivariant velocity field $\hat{v}$ started from symmetric initial configurations (first column). The intermediate configurations are related by the same initial transformation, illustrating Proposition 8.

**Proposition 22.** *Given a fixed composition $\{n_s\}_{s \in \mathcal{S}}$, the canonical distribution associated to $p_\star$ is*

$$p_\star^n(\mathrm{d}s, \mathrm{d}vol_X) \;=\; \frac{1}{\mathcal{Z}^n} \exp\left(-\frac{\mathrm{U}_\star(s, X)}{k_{\mathbf{B}}T}\right) \times \prod_{s' \in \mathcal{S}} \delta_{\sum_{i=1}^N \delta_{s'}(s_i)}(n_{s'}) \, \mathrm{d}s \, \mathrm{d}vol_X \;.$$

*This density is $G_{\mathcal{C}}$-invariant.*

*Proof.* Use the $G_{\mathcal{C}}$-invariance of $p_\star$ and the permutation invariance of the sum. $\qquad\square$

**Proposition 23.** *Let $A, B \in \mathbb{R}^d$, $M \in B_d$ and $u \in \mathbb{R}^d$ then*

$$\exp_{(MA+u) \,\%\, L}(MB) = (M \exp_A B + u) \,\%\, L$$

*holds.*

*Proof.* Using Lemma 11, we have that

$$\exp_{(MA+u) \,\%\, L}(MB) = \left(\exp_{(MA) \,\%\, L} MB + u\right) \,\%\, L \;.$$

Moreover, given that the signed permutation matrix $M$ is characterized by permutation $\sigma$ and a vector $\epsilon \in \{-1, 1\}^d$, we have for all $i \in [\![1, d]\!]$

$$[M(A+B)]_i = \epsilon_i(A_{\sigma(i)} + B_{\sigma(i)}) \;.$$

Using Lemmas 11 and 12, we get that

$$\left[\exp_{(MA) \,\%\, L} MB\right]_i = \left(\epsilon_i(A_{\sigma(i)} + B_{\sigma(i)})\right) \,\%\, L = \left(\epsilon_i\left[(A_{\sigma(i)} + B_{\sigma(i)}) \,\%\, L\right]\right) \,\%\, L \;.$$

Additionally, note that

$$[(M \exp_A B) \,\%\, L]_i = \left(\epsilon_i\left[(A_{\sigma(i)} + B_{\sigma(i)}) \,\%\, L\right]\right) \,\%\, L \;,$$

which proves that $\exp_{(MA) \,\%\, L} MB = (M \exp_A B) \,\%\, L$. If we put this expression in the first equation and use Lemma 11, we get

$$\exp_{(MA+u) \,\%\, L}(MB) = ((M \exp_A B) \,\%\, L + u) \,\%\, L = (M \exp_A B + u) \,\%\, L \;.$$

$\qquad\square$

## B    RIEMANNIAN STOCHASTIC INTERPOLANTS EQUIVARIANCE

**Lemma 24.** *$G_{\mathcal{C}}$ is a group for the composition operation.*

*Proof.* The identity is in $G_{\mathcal{C}}$ as it is a special case of the invariances. Let $g^1, g^2 \in G_{\mathcal{C}}$ we can decompose them as $g^1 = f_L \circ h_{A^1, b^1}$ and $g^2 = f_L \circ h_{A^2, b^2}$ where $A_1$ (respectively $A^2$) can be decomposed into $A_\sigma^1$, $M^1$ (respectively $A_\sigma^2$, $M^2$) and $b^1$ (respectively $b^2$) into $b^1 = \mathbf{1}_N \otimes c^1$ (respectively $b^2 = \mathbf{1}_N \otimes c^2$). Using both Lemmas 13 and 11, $g^1 \circ g^2$ can be written for any $C \in \mathcal{C}$

$$(g^1 \circ g^2)(C) = f_L \left[ \begin{pmatrix} A_{\sigma^1} A_{\sigma^2} & 0 \\ 0 & (A_{\sigma^1} A_{\sigma^2} \otimes I_d)(I_N \otimes M^1 M^2) \end{pmatrix} C + (\mathbf{1}_N \otimes (c^1 + M^1 c^2)) \right] \ ,$$

which shows, by Lemma 10, that $g^1 \circ g^2 \in G_{\mathcal{C}}$. Finally, let $g \in G_{\mathcal{C}}$ be written under the same decomposition, using Lemmas 13 and 11 again, we have for any $C \in \mathcal{C}$

$$g^{-1}(C) = f_L \left[ \begin{pmatrix} A_{\sigma^{-1}} & 0 \\ 0 & (I_N \otimes M^{-1})(A_{\sigma^{-1}} \otimes I_d) \end{pmatrix} C - (\mathbf{1}_N \otimes c) \right] \ ,$$

$$= f_L \left[ \begin{pmatrix} A_{\sigma^{-1}} & 0 \\ 0 & (A_{\sigma^{-1}} \otimes I_d)(I_N \otimes M^{-1}) \end{pmatrix} C + (\mathbf{1}_N \otimes (-c)) \right] \ .$$

which shows, by Lemma 10, that $g^{-1} \in G_{\mathcal{C}}$. $\qquad \square$

**Lemma 25.** *For all $g = f_L \circ h_{A,b} \in G_{\mathcal{C}}$, as in Equation (4), let $J_g$ be the Jacobian of $g$. Then $J_g(C) = A$, for almost every $C \in \mathcal{C}$, and $|J_g(C)| = 1$ almost everywhere.*

*Proof.* Let $C \in \mathcal{C}$, then by the chain rule

$$J_g(C) = J_{f_L}\left( h_{A,b}(C) \right) J_{h_{A,b}}(C) \ .$$

The modulo has unit derivative (except from jump points), so $J_{f_L} = I_{(N+1)d}$ almost everywhere. Since $h_{A,b}$ is affine, $J_{h_{A,b}} = A$, so $J_g(C) = A$, for almost every $C \in \mathcal{C}$. Moreover, $|J_g(C)| = |A|$. Using Equation (15), we can show that $A \in \mathcal{B}_{Nd}$ which implies that $|A| = 1$. $\qquad \square$

**Lemma 26.** *Let $g \in G_{\mathcal{C}}$ and $C, D \in \mathcal{C}$ and then*

$$\int_{\mathcal{C}} \varphi(C) \delta_{g(D)}\left( g(C) \right) \mathrm{d}C = \int_{\mathcal{C}} \varphi(C) \delta_D\left( C \right) \mathrm{d}C \ ,$$

*for all $\varphi \in C^\infty(\mathcal{C})$.*

*Proof.* Make the change of variable $C = g(U)$, since $g$ is a group element,

$$U = g^{-1}(C), \quad \mathrm{d}C = \left| J_{g^{-1}}(U) \right| \mathrm{d}U \ .$$

Using Lemma 25, $\left| J_{g^{-1}}(U) \right| = 1$, thus

$$\int_{\mathcal{C}} \varphi(C) \delta_{g(D)}\left( g(C) \right) \mathrm{d}C = \int_{\mathcal{C}} \varphi\left( g^{-1}(U) \right) \delta_{g(D)}\left( U \right) \mathrm{d}U = \varphi(D) = \int_{\mathcal{C}} \varphi(C) \delta_D(C) \mathrm{d}C \ .$$

$\qquad \square$

**Proposition 27.** *Given a base $p_{\mathrm{base}}$ and the target $p_\star$ distribution both $G_{\mathcal{C}}$-invariant and a $G_{\mathcal{C}}$-equivariant interpolant $I$, then the induced marginal densities $(p_t)_{t=0}^1$ are all $G_{\mathcal{C}}$-invariant.*

*Proof.* The marginal densities are defined as

$$p_t(C) = \int_{\mathcal{C}} \delta_{I(t, C_0, C_1)}(C)\, p_{\mathrm{base}}(C_0)\, p_\star(C_1)\, \mathrm{d}C_0 \mathrm{d}C_1 \ .$$

Let $g \in G_{\mathcal{C}}$, then

$$p_t(g(C)) = \int_{\mathcal{C}} \delta_{I(t, C_0, C_1)}(g(C))\, p_{\mathrm{base}}(C_0)\, p_\star(C_1)\, \mathrm{d}C_0 \mathrm{d}C_1 \ .$$

Make the change of variable $C_0 = g(U_0)$, $C_1 = g(U_1)$, using Lemma 25 we get

$$p_t(g(C)) = \int_{\mathcal{C}} \delta_{I(t,g(U_0),g(U_1))}(g(C)) \, p_{\text{base}}(g(U_0)) \, p_\star(g(U_1)) \, \mathrm{d}U_0 \mathrm{d}U_1 \ ,$$

$$= \int_{\mathcal{C}} \delta_{I(t,g(U_0),g(U_1))}(g(C)) \, p_{\text{base}}(U_0) \, p_\star(U_1) \, \mathrm{d}U_0 \mathrm{d}U_1 \ , \tag{19}$$

$$= \int_{\mathcal{C}} \delta_{g(I(t,U_0,U_1))}(g(C)) \, p_{\text{base}}(U_0) \, p_\star(U_1) \, \mathrm{d}U_0 \mathrm{d}U_1 \ , \tag{20}$$

$$= \int_{\mathcal{C}} \delta_{I(t,U_0,U_1)}(C) \, p_{\text{base}}(U_0) \, p_\star(U_1) \, \mathrm{d}U_0 \mathrm{d}U_1 \ , \tag{21}$$

$$= p_t(C) \ .$$

We used the invariance of $p_{\text{base}}$ and $p_\star$ to get (19), the equivariance of $I$ to go from (19) to (20) and Lemma 26 to go from (20) to (21). $\qquad\square$

**Proposition 28.** *The interpolant in Equation* (10) *is $G_{\mathcal{C}}$-equivariant.*

*Proof.* Let $g \in G_{\mathcal{C}}$. Consider the decomposition of $g$ introduced in Lemma 10. If we denote

$$I_L : (t, C_0, C_1) \mapsto \begin{pmatrix} I_L^s(t, C_0, C_1) \\ I_L^X(t, C_0, C_1) \end{pmatrix} \ ,$$

then

$$I_L^s(t, g(C_0), g(C_1)) = (1-t)A_\sigma s_0 + t A_\sigma s_1 = A_\sigma I_L^s(t, C_0, C_1) \ .$$

Moreover, for every $i \in [\![1, N]\!]$, by definition

$$I_L^X(t, g(C_0), g(C_1))_{(i)} = \exp_{\tilde{h}_{M,c,L}((X_0)_{(\sigma(i))})} \left( t \log_{\tilde{h}_{M,c,L}((X_0)_{(\sigma(i))})} \tilde{h}_{M,c,L}((X_1)_{(\sigma(i))}) \right) \ ,$$

$$= \exp_{\tilde{h}_{M,c,L}((X_0)_{(\sigma(i))})} \left( tM \log_{(X_0)_{(\sigma(i))}}(X_1)_{(\sigma(i))} \right) \ ,$$

$$= \left( M \exp_{(X_0)_{(\sigma(i))}} \left( t \log_{(X_0)_{(\sigma(i))}}(X_1)_{(\sigma(i))} \right) + c \right) \ \% \ L$$

$$= \tilde{h}_{M,c,L}(I^X(t, C_0, C_1))_{(\sigma(i))}$$

where we used successively Corollary 20 and Proposition 23, which concludes the proof. $\qquad\square$

## C  MODEL'S EQUIVARIANCE

**Lemma 29.** *Let $g \in G_{\mathcal{C}}$ and $T : \mathcal{C} \to \mathcal{C}$ be a diffeomorphism. $T$ is $G_{\mathcal{C}}$-equivariant if and only if $T^{-1}$ is $G_{\mathcal{C}}$-equivariant.*

*Proof.* Let $Y \in \mathcal{C}$ and set $X = T(Y)$, then

$$T^{-1}(g(X)) = g(T^{-1}(X)) \iff T^{-1}(g(T(Y))) = g(Y)$$

$$\iff g(T(Y)) = T(g(Y)) \ .$$

$\qquad\square$

The following proposition generalizes (Köhler et al., 2020, Theorem 1) to $G_{\mathcal{C}}$ which is nonlinear.

**Proposition 30.** *Let $q : \mathcal{C} \to \mathbb{R}$ be a density of $\mathcal{C}$ and let $T : \mathcal{C} \to \mathcal{C}$ be a diffeomorphism. If $q$ is $G_{\mathcal{C}}$-invariant and $T$ is a $G_{\mathcal{C}}$-equivariant, then the push-forward of $q$ through $T$ is also $G_{\mathcal{C}}$-invariant.*

*Proof.* Let $C \in \mathcal{C}$, the push-forward of $q$ through $T$ writes as

$$p(C) = q\left(T^{-1}(C)\right) |J_{T^{-1}}(C)| \ .$$

Thus

$$p(g(C)) = q\left(T^{-1}(g(C))\right) |J_{T^{-1}}(g(C))| \ .$$

By Lemma 29 and the invariance of $q$ we have

$$q\left(T^{-1}\left(g(C)\right)\right) = q\left(g\left(T^{-1}\left(C\right)\right)\right)$$
$$= q\left(T^{-1}(C)\right).$$

Moreover,

$$
\begin{aligned}
\left|J_{T^{-1}}\left(g(C)\right)\right| &= \frac{\left|J_{T^{-1}\circ g}(C)\right|}{\left|J_g(C)\right|} & \text{(Chain rule)} \\
&= \left|J_{T^{-1}\circ g}(C)\right| & \text{(Lemma 25)} \\
&= \left|J_{g\circ T^{-1}}(C)\right| & \text{(Lemma 29)} \\
&= \left|J_g\left(T^{-1}(C)\right)\right|\left|J_{T^{-1}}(C)\right| & \text{(Chain rule)} \\
&= \left|J_{T^{-1}}(C)\right| & \text{(Lemma 25)}.
\end{aligned}
$$

Combining these results we get $p\left(g(C)\right) = p(C)$. $\qquad\square$

**Proposition 31.** *If $v : [0,1] \times \mathcal{C} \to \mathcal{TC}$ is a Lipschitz-bounded $G_{\mathcal{C}}$-equivariant velocity field, then the transport map induced by its ODE flow is $G_{\mathcal{C}}$-equivariant.*

*Proof.* Let $g \in G_{\mathcal{C}}$ be decomposed (as per Lemma 10) as $g = f_L \circ h_{A,b}$. Let $Z \in \mathcal{C}$. Consider $C(\cdot, Z)$ denotes the solution of the ODE with initial condition $Z \in \mathcal{C}$. Let $\tilde{C}(\cdot, Z) = C(\cdot, g(Z))$. Using the chain rule and the equivariance of $v$, we have almost everywhere that

$$\frac{\mathrm{d}\tilde{C}(t,Z)}{\mathrm{d}t} = A\frac{\mathrm{d}C(t,Z)}{\mathrm{d}t} = Av(t, C(t,Z)) = v(t, g(C(t,Z))) = v(t, \tilde{C}(t,Z)).$$

Moreover, $\tilde{C}(0,Z) = g(Z)$. Using the unicity of the ODE's solutions (due to the velocity field being Lipschitz-bounded), we get

$$C(t, g(Z)) = g(C(t,Z)), \quad \text{almost everywhere, for all } t \in [0,1].$$

Using this into the integral definition of the transport map, for all $t \in [0,1]$ we get

$$
\begin{aligned}
T(g(Z)) &= f_L\left(g(Z) + \int_0^t v(u, C(t, g(Z)))\mathrm{d}u\right) \\
&= f_L\left(g(Z) + \int_0^t v(u, g(C(t,Z)))\mathrm{d}u\right) & \text{(Using } C(\cdot, g(Z)) = g(C(\cdot, Z))) \\
&= f_L\left(f_L(h_{A,b}(Z)) + A\int_0^t v(u, C(t,Z))\mathrm{d}u\right) & \text{(Using the equivariance of } v) \\
&= f_L\left(h_{A,b}(Z) + A\int_0^t v(u, C(t,Z))\mathrm{d}u\right) & \text{(Using Lemma 11)} \\
&= f_L\left(h_{A,b}\left(Z + \int_0^t v(u, C(t,Z))\mathrm{d}u\right)\right) \\
&= g(T(Z)).
\end{aligned}
$$

$\qquad\square$

**Lemma 32.** *Let $g \in G_{\mathcal{C}}$ and $I : [0,1] \times \mathcal{C} \times \mathcal{C} \to \mathcal{C}$ be a differentiable, $G_{\mathcal{C}}$-equivariant interpolation function. Then, the time derivative $\partial_t I$ is $\mathcal{C}$-equivariant, i.e., it verifies for any $t \in [0,1]$, $C_0, C_1 \in \mathcal{C}$*

$$\partial_t I(t, g(C_0), g(C_1)) = A\partial_t I(t, C_0, C_1), \quad \text{for all } g \in G_{\mathcal{C}} \text{ such that } g = f_L \circ h_{A,b}.$$

*Proof.* Let $C_0, C_1 \in \mathcal{C}$, then

$$
\begin{aligned}
\partial_t I\left(t, g\left(C_0\right), g\left(C_1\right)\right) &= \partial_t g\left(I\left(t, C_0, C_1\right)\right) & \text{(Equivariance of } I) \\
&= J_g\left(I\left(t, C_0, C_1\right)\right)\partial_t I\left(t, C_0, C_1\right) & \text{(Chain rule)} \\
&= A\partial_t I\left(t, C_0, C_1\right) & \text{(Lemma 25)},
\end{aligned}
$$

where $A$ is defined in Equation (4). $\qquad\square$

**Proposition 33.** *Given a $G_{\mathcal{C}}$-equivariant interpolation function $I$, if $p_{base}$ and $p_{\star}$ are $G_{\mathcal{C}}$-invariant, then the corresponding optimal velocity field is $G_{\mathcal{C}}$-equivariant.*

*Proof.* The optimal velocity field is defined as

$$u^{\star}(t, C) = \frac{1}{p_t(C)} \int_{\mathcal{C}} \partial_t I(t, C_0, C_1) \, \delta_{I(t,C_0,C_1)}(C) p_{\text{base}}(C_0) \, p_{\star}(C_1) \, \mathrm{d}C_0 \mathrm{d}C_1 \;.$$

Since $p_t$ is $G_{\mathcal{C}}$-invariant (see Proposition 4), we need to show that the integral

$$\mathcal{I}(C) = \int_{\mathcal{C}} \partial_t I(t, C_0, C_1) \, \delta_{I(t,C_0,C_1)}(C) p_{\text{base}}(C_0) \, p_{\star}(C_1) \, \mathrm{d}C_0 \mathrm{d}C_1$$

is a $G_{\mathcal{C}}$-equivariant vector field. Let $g \in G_{\mathcal{C}}$,

$$\mathcal{I}(g(C)) = \int_{\mathcal{C}} \partial_t I(t, C_0, C_1) \, \delta_{I(t,C_0,C_1)}(g(C)) \, p_{\text{base}}(C_0) \, p_{\star}(C_1) \, \mathrm{d}C_0 \mathrm{d}C_1 \;.$$

Make the change of variable $C_0 = g(U_0)$, $C_1 = g(U_1)$, using Lemma 25,

$$\mathcal{I}(g(C)) = \int_{\mathcal{C}} \partial_t I(t, g(U_0), g(U_1)) \, \delta_{I(t,g(U_0),g(U_1))}(g(C)) \, p_{\text{base}}(g(U_0)) \, p_{\star}(g(U_1)) \, \mathrm{d}U_0 \mathrm{d}U_1$$

$$= \int_{\mathcal{C}} \partial_t I(t, g(U_0), g(U_1)) \, \delta_{I(t,g(U_0),g(U_1))}(g(C)) \, p_{\text{base}}(U_0) \, p_{\star}(U_1) \, \mathrm{d}U_0 \mathrm{d}U_1 \tag{22}$$

$$= \int_{\mathcal{C}} A \partial_t I(t, U_0, U_1) \, \delta_{g(I(t,U_0,U_1))}(g(C)) \, p_{\text{base}}(U_0) \, p_{\star}(U_1) \, \mathrm{d}U_0 \mathrm{d}U_1 \tag{23}$$

$$= \int_{\mathcal{C}} A \partial_t I(t, U_0, U_1) \, \delta_{I(t,U_0,U_1)}(C) \, p_{\text{base}}(U_0) \, p_{\star}(U_1) \, \mathrm{d}U_0 \mathrm{d}U_1 \tag{24}$$

$$= A\mathcal{I}(C) \;,$$

where we use the invariance of $p_{\text{base}}$ and $p_{\star}$ in (22), then the equivariance of $\partial_t I$ due to Lemma 32 to go from (22) to (23) and Lemma 26 to go from (23) to (24). □

## D GRAPH NEURAL NETWORK EQUIVARIANCE

**Proposition 34.** *The velocity field in Equation* (14) *is $G_{\mathcal{C}}$-equivariant.*

*Proof.* Let $t \in [0, 1]$ and $C \in \mathcal{C}$. We define

$$\psi_k(t, C) = (\psi_k^s(t, C), \psi_k^X(t, C)) = (s, X^k), \ \ \Gamma_k(t, C) = H^k \text{ and } \Lambda_k(t, C) = M^k \;.$$

We start by showing by induction on $k \in \mathbb{N}$ that for all $g \in G_{\mathcal{C}}$ and $i, j \in [\![1, N]\!]$, we have

$$\psi_k(t, g(C)) = g(\psi_k(t, C)), \ [\Gamma_k(t, g(C))]_i = [\Gamma_k(t, C)]_{\sigma(i)} \text{ and } \ [\Lambda_k(t, g(C))]_{i,j} = [\Lambda_k(t, C)]_{\sigma(i),\sigma(j)} \;,$$

where, building on Equation (4), we decompose $g$ as

$$g : \begin{pmatrix} s \\ X \end{pmatrix} \mapsto \begin{pmatrix} g^s(C) \\ g^X(C) \end{pmatrix} = \begin{pmatrix} A_{\sigma} s \\ (A_{\sigma} \otimes I_d) h_{M,b,L}(X) \end{pmatrix} \;,$$

$$h_{M,b,L}(X) = ((I_N \otimes M)X + (\mathbf{1}_N \otimes b)) \ \% \ L \;,$$

where $h_{M,b,L}$ describes the group of invariances restricted to $\mathcal{M}^N$ (denoted $\mathcal{G}_{\mathcal{M}^N}$), $\sigma \in S_N$ is a permutation, $A_{\sigma}$ is the associated permutation matrix of size $N \times N$, $M$ is a signed permutation matrix of size $d \times d$ and $b$ is a vector in $\mathbb{R}^d$.

At $k = 0$, we have $\psi_0(t, g(C)) = g(C) = g(\psi_0(t, C))$ and for all $i \in [\![1, N]\!]$

$$[\Gamma_0(t, g(C))]_i = (t, [g^s(C)]_i) = (t, s_{\sigma(i)}) = [\Gamma_0(t, C)]_{\sigma(i)} \;.$$

Moreover, for all $i, j \in [\![1, N]\!]$

$$[\Lambda_0(t, g(C))]_{i,j} = \hat{\phi}_e \left( [\Gamma_0(t, g(C))]_i, [\Gamma_0(t, g(C))]_j, \mathrm{d}_{\mathcal{M}}(h_{M,b,L}(X)_{(\sigma(i))}, h_{M,b,L}(X)_{(\sigma(j))}) \right) \;.$$

Using the previous statement on $\Gamma_0$ as well as the invariance of $\mathrm{d}_{\mathcal{M}}$ to $\mathcal{G}_{\mathcal{M}}$ (see Corollary 21),

$$[\Lambda_0(t, g(C))]_{i,j} = \hat{\phi}_e\left([\Gamma_0(t, C)]_{\sigma(i)}, [\Gamma_0(t, C)]_{\sigma(j)}, \mathrm{d}_{\mathcal{M}}(X_{(\sigma(i))}, X_{(\sigma(j))})\right),$$

$$= [\Lambda_k(t, g(C))]_{\sigma(i), \sigma(j)}.$$

Let $k \in \mathbb{N}$ and assume that

$$\psi_k(t, g(C)) = g(\psi_k(t, C)), \ [\Gamma_k(t, g(C))]_i = [\Gamma_k(t, C)]_{\sigma(i)} \ \text{and} \ [\Lambda_k(t, g(C))]_{i,j} = [\Lambda_k(t, C)]_{\sigma(i), \sigma(j)},$$

Let $i, j \in [\![1, N]\!]$, using the recursion assumption and the permutation invariance of the sum, we have that

$$[\Gamma_{k+1}(t, g(C))]_i = \hat{\phi}_h\left([\Gamma_k(t, g(C))]_i, \sum_{i \neq j} \hat{\phi}_m\left([\psi_k(t, g(C))]_{i,j}\right)[\psi_k(t, g(C))]_{i,j}\right),$$

$$= \hat{\phi}_h\left([\Gamma_k(t, C)]_{\sigma(i)}, \sum_{i \neq j} \hat{\phi}_m\left([\psi_k(t, C)]_{\sigma(i), \sigma(j)}\right)[\psi_k(t, C)]_{\sigma(i), \sigma(j)}\right),$$

$$= \hat{\phi}_h\left([\Gamma_k(t, C)]_{\sigma(i)}, \sum_{i \neq j} \hat{\phi}_m\left([\psi_k(t, C)]_{\sigma(i), j}\right)[\psi_k(t, C)]_{\sigma(i), j}\right),$$

$$= [\Gamma_{k+1}(t, C)]_{\sigma(i)}.$$

Similarly, using the same argument as in $k = 0$, it is easy to show that

$$[\Lambda_{k+1}(t, g(C))]_{i,j} = [\Lambda_k(t, C)]_{\sigma(i), \sigma(j)}.$$

Additionally, we have

$$\psi_{k+1}^s(t, g(C)) = g^s(C) = g^s(\psi_k^s(t, C)).$$

Moreover, using the definition and the previous statement on $\Lambda_k$

$$\psi_{k+1}^X(t, g(C))_{(i)} = \exp_{\psi_k^X(t, g(C))_{(i)}}\left(\sum_{i \neq j} \frac{\log_{\psi_k^X(t, g(C))_{(j)}} \psi_k^X(t, g(C))_{(i)}}{\mathrm{d}_{\mathcal{M}}(g^X(C)_{(i)}, g^X(C)_{(j)}) + 1} \phi_d([\Lambda_k(t, g(C))]_{i,j})\right),$$

$$= \exp_{\psi_k^X(t, g(C))_{(i)}}\left(\sum_{i \neq j} \frac{\log_{\psi_k^X(t, g(C))_{(j)}} \psi_k^X(t, g(C))_{(i)}}{\mathrm{d}_{\mathcal{M}}(g^X(C)_{(i)}, g^X(C)_{(j)}) + 1} \phi_d([\Lambda_k(t, C)]_{\sigma(i), \sigma(j)})\right).$$

Using the recursion assumption together with Corollary 20 we have

$$\log_{\psi_k^X(t, g(C))_{(j)}} \psi_k^X(t, g(C))_{(i)} = \log_{g^X(\psi_k(t, C))_{(j)}} g^X(\psi_k(t, C))_{(i)},$$

$$= \log_{h_{M,b,L}(\psi_k^X(t, C))_{(\sigma(j))}} h_{M,b,L}(\psi_k^X(t, C))_{(\sigma(i))},$$

$$= M \log_{\psi_k^X(t, C)_{(\sigma(j))}} \psi_k^X(t, C)_{(\sigma(i))}.$$

Using Proposition 23 and the recursion assumption again, we get that

$$\psi_{k+1}^X(t, g(C))_{(i)} = \left(M \exp_{\psi_k^X(t, C)_{(\sigma(i))}} \sum_{i \neq j} \frac{\log_{\psi_k^X(t, C)_{(\sigma(j))}} \psi_k^X(t, C)_{(\sigma(i))}}{\mathrm{d}_{\mathcal{M}}(g^X(C)_{(i)}, g^X(C)_{(j)}) + 1} \phi_d([\Lambda_k(t, C)]_{\sigma(i), \sigma(j)}) + b\right) \% L.$$

Together with the invariance of $\mathrm{d}_{\mathcal{M}}$ and the permutation invariance of the sum, it leads to

$$\psi_{k+1}^X(t, g(C))_{(i)} = \left(M \exp_{\psi_k^X(t, C)_{(\sigma(i))}} \sum_{i \neq j} \frac{\log_{\psi_k^X(t, C)_{(j)}} \psi_k^X(t, C)_{(\sigma(i))}}{\mathrm{d}_{\mathcal{M}}(X_{(\sigma(i))}, X_{(j)}) + 1} \phi_d([\Lambda_k(t, C)]_{\sigma(i), j}) + b\right) \% L,$$

$$= \left[\left(M \psi_{k+1}^X(t, g(C))_{(i)} + b\right) \% L\right]_{(\sigma(i))},$$

$$= g^X(\psi_{k+1}^X(t, C))_{(i)},$$

which concludes the recursive proof. We get the overall proof using Corollary 20 as

$$
\begin{aligned}
\hat{v}(t, g(C)) &= \begin{pmatrix} \mathbf{0}_{N \times d_s} \\ \left\{ \log_{g^X(X)_{(i)}} \psi_K^X(t, g(C))_{(i)} \right\}_{i=1}^N \end{pmatrix} , \\
&= \begin{pmatrix} \mathbf{0}_{N \times d_s} \\ \left\{ \log_{g^X(X)_{(i)}} g^X(\psi_K^X(t, C))_{(i)} \right\}_{i=1}^N \end{pmatrix} , & \text{(Recursion)} \\
&= \begin{pmatrix} \mathbf{0}_{N \times d_s} \\ \left\{ M \log_{X_{(\sigma(i))}} \psi_K^X(t, C)_{(\sigma(i))} \right\}_{i=1}^N \end{pmatrix} , & \text{(Corollary 20)} \\
&= A\hat{v}(t, C)
\end{aligned}
$$

$\square$

## E   IMPORTANCE SAMPLING

Importance Sampling (IS) estimates expectations under a target distribution $\pi$ using a proposal distribution $\rho$ that is easy to sample from. Assuming $\mathrm{supp}(\pi) \subseteq \mathrm{supp}(\rho)$, one can write

$$
\mathbb{E}_\pi[\phi(Y)] = \mathbb{E}_\rho \left[ \phi(Y) \frac{\pi(Y)}{\rho(Y)} \right] .
$$

This motivates the Monte Carlo estimator

$$
\hat{\pi}_\phi^N = \frac{1}{N} \sum_{i=1}^N \phi(Y_i) \, w(Y_i), \quad w(Y_i) = \frac{\pi(Y_i)}{\rho(Y_i)}, \quad Y_i \sim \rho \;\; i.i.d.
$$

where $w(Y_i)$ are the importance weights and $\{Y_i\}_{i=1}^N$ the particle set. In practice, $\pi$ is often known only up to a normalizing constant, making $w(Y_i)$ intractable. Self-normalized importance sampling addresses this via

$$
\bar{\pi}_\phi^N = \sum_{i=1}^N \phi(Y_i) \frac{w(Y_i)}{\sum_{j=1}^N w(Y_j)} ,
$$

which is biased but consistent as $N \to \infty$. The Effective Sample Size (ESS) quantifies how many independent samples from the target distribution $\pi$ would provide the same statistical efficiency as the weighted particle set $Y_i, w(Y_i)$. A common estimator is

$$
\mathrm{ESS}(Y_{1:N}) = \frac{1}{\sum_{i=1}^N \bar{w}(Y_i)^2}, \quad \bar{w}(Y_i) = \frac{w(Y_i)}{\sum_{j=1}^N w(Y_j)} ,
$$

which decreases when the weights are highly imbalanced, indicating that only a few particles effectively contribute to the estimate.

## F   EXPERIMENTAL DETAILS

### F.1   GENERATION OF THE TRAINING DATASET.

We generated the training datasets with the Metropolis–Hastings Monte Carlo algorithm (Metropolis et al., 1953). Starting from particles uniformly distributed in the box, the update kernel consists in selecting one particle randomly with equal probability and attempting a displacement drawn from a Gaussian distribution centred at the current position with standard deviation 0.065 (Frenkel & Smit, 2002). The move is then accepted or rejected according to the standard Metropolis criterion. We define one unit of time as $N$ attempted moves. We ran 100 independent chains, each initialised from a different random configuration, for $10^4$ time units to reach equilibrium. In this time scale, the potential energy rapidly relaxes to a steady value, and the self-intermediate scattering function (Berthier & Biroli, 2011) – the standard time-correlation function used to quantify structural relaxation in liquids – decays to zero within $10^4$ time units, confirming that the system is equilibrated. From these equilibrated configurations, each chain was then propagated for an additional $10^7$ time units, storing one configuration every $10^4$ steps. This procedure yields a total of $10^5$ uncorrelated equilibrium configurations, which we use as training data.

Table 1: Chosen learning rates (LR) and gradient clipping (GC) for each model and system size. Architectures are described by the number of GNN layers $K$ and hidden feature size (HF). Unless specified otherwise, GC is applied with a threshold of 2.0.

| System | Model | Architecture ($K \mid$ HF) | LR (GC) |
|--------|-------|----------------------------|---------|
| | eRSI | $3 \mid 32$ | $5 \times 10^{-3}$ |
| | eRSI | $4 \mid 64$ | $1 \times 10^{-3}$ |
| $N = 44$ | eRSI | $5 \mid 128$ | $1 \times 10^{-3}$ |
| | eFM | $3 \mid 32$ | $1 \times 10^{-3}$ |
| | RSI | $-$ | $5 \times 10^{-5}$ |
| | eRSI | $3 \mid 32$ | $1 \times 10^{-4}$ |
| | eRSI | $4 \mid 64$ | $5 \times 10^{-5}$ (no GC) |
| $N = 10$ | eRSI | $5 \mid 128$ | $1 \times 10^{-5}$ |
| | eFM | $3 \mid 32$ | $5 \times 10^{-3}$ |
| | RSI | $-$ | $5 \times 10^{-4}$ |

eRSI model samples                                      Dataset samples

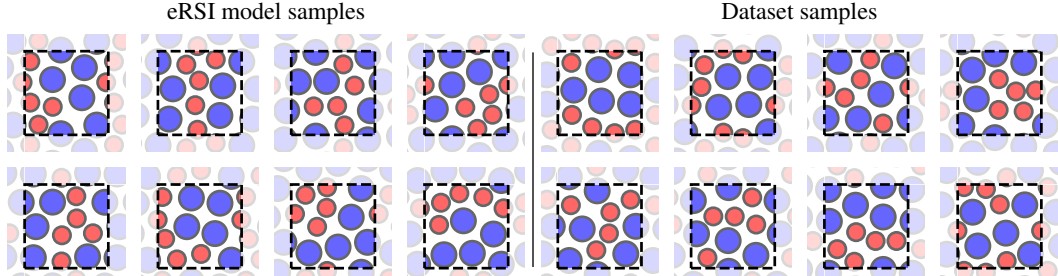

Figure 6: **Additional samples for the IPL system with** $N = 10$**.** Configurations generated by the eRSI model *(left)* compared with reference samples from the dataset *(right)*.

### F.2 DESIGNS OF THE VELOCITY FIELDS

In the GNN implementation (14), we set $K = 3$. The networks $\hat{\phi}_e$ and $\hat{\phi}_h$ are 3-layer MLPs with width 32, while $\hat{\phi}_m$ is a 2-layer MLP with width 32. Species are embedded using one-hot encoding. The network has a total of approximately 22k trainable parameters. For eFM, we adopt the GNN architecture of Satorras et al. (2021), corresponding to Equation (14) but with Euclidean operations: $\exp_x(y) = x + y$, $\log_x(y) = y - x$, and the standard Euclidean distance. The same hyperparameters as in eRSI are used. For RSI, the configuration is fed directly into a 3-layer MLP with 64 hidden units per layer, yielding a comparable parameter count to the GNN models. To account for the torus geometry, inputs are projected onto the manifold following (Chen & Lipman, 2024, Equation 26).

### F.3 TRAINING AND SAMPLING DETAILS

Models were trained for 1000 epochs on datasets of size $10^5$, with batch sizes of 2048 samples for IPL with $N = 10$ and 1024 for $N = 44$. Training used the Adam optimizer, with hyperparameters selected via validation loss among learning rates $\{10^{-5}, 5 \times 10^{-5}, 10^{-4}, 5 \times 10^{-4}, 10^{-3}, 5 \times 10^{-3}\}$ and with or without gradient clipping at 2.0. The choices are recapped in Table 1. For the non-equivariant RSI baseline, additional data augmentation using random group actions of $G_{\mathcal{C}}$ was tested but yielded no improvement.

Sampling follows the guidelines of Chen et al. (2019), using the Dormand–Prince–Shampine Runge–Kutta method of order 5 (`dopri5`) from `torchdiffeq` (Chen, 2018), with absolute and relative tolerances set to $10^{-5}$. Divergences in likelihood computations (see Equation (8)) are evaluated exactly via automatic differentiation.

### F.4 ADDITIONAL RESULTS

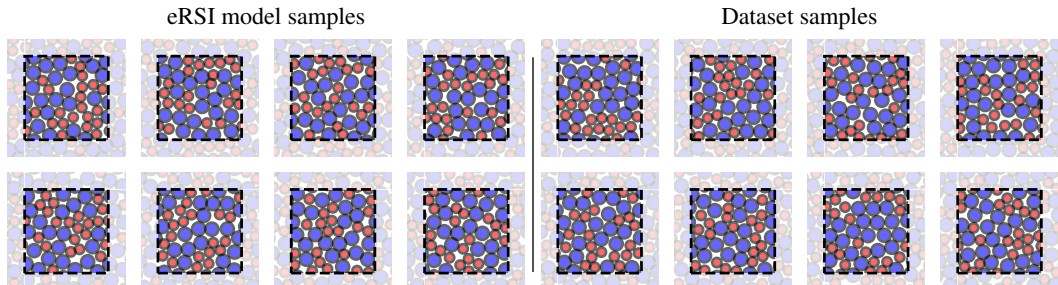

eRSI model samples                    Dataset samples

Figure 7: **Additional samples for the IPL system with** $N = 44$**.** Configurations generated by the eRSI model *(left)* compared with reference samples from the dataset *(right)*.

**Additional samples from eRSI**   Additional samples from both the reference models and the dataset are shown in Figures 6 and 7.

**Effective sample size and energy overlap.**   A central difficulty in reweighting is that the ESS (see Appendix E) can be extremely low in high dimensions. In our experiments, we observe that the ESS decays approximately as $1/R$ over a broad range of sample sizes $R$, before eventually stabilising to a finite plateau only at large $R$. This behaviour implies that at small $R$, the ESS is essentially uninformative: estimating it reliably requires a number of samples that may itself exceed the plateau value. For $N = 10$, Figure 8(a–c) shows that the crossover scale $\overline{R}$ at which the ESS departs from the $1/R$ trend is strongly model-dependent. For RSI, no such crossover is observed: the ESS remains proportional to $1/R$ throughout. For eFM, $\overline{R} \approx 10^4$, while for eRSI it is significantly smaller, $\overline{R} \approx 10^3$. To explain this phenomenon, we compare the energy distributions of samples from the three models with that of the target, shown in Figure 8(d–f). For clarity, we discard configurations with energies larger than twice the maximum observed in the target dataset. The discarded fraction is 100% for RSI, about 84% for eFM, and about 3% for eRSI. For small $R$, the two distributions $p(U)$ (target) and $q(U)$ (model) overlap poorly: $q(U)$ is concentrated at higher energies and only begins to penetrate the region of significant $p(U)$ weight from the right tail. As $R$ increases, rare samples from this region appear with very small $q(U)$ but relatively large $p(U)$, producing very large reweighting factors. A few such configurations dominate the averages, driving the $1/R$ behaviour of the ESS. Once $R$ exceeds $\overline{R}$, $q(U)$ has infiltrated sufficiently deep into the bulk of $p(U)$ so that ratios $p(U)/q(U)$ are less extreme, and the ESS stabilises to a plateau. We repeat the analysis for $N = 44$ in Figure 9, using up to $R = 8 \times 10^3$ samples for RSI and eFM and up to $R = 10^6$ samples for eRSI. The results are qualitatively the same, although eFM samples are significantly worse due to overlaps at the boundaries (see Figure 2).

**Network size dependence.**   Figure 10 shows the effect of network size on the IPL system with $N = 44$. Larger models ($\approx$580k parameters vs. 22k for the smallest) achieve substantially higher fidelity and are expected to reweight observables with far fewer samples. This gain, however, comes at the cost of a $\sim$2.5$\times$ slower training and a $\sim$10$\times$ slower sampling on an NVIDIA A100 GPU.

**Dataset size dependence.**   Figure 11 shows that increasing the training dataset size consistently improves model quality, with larger datasets yielding energy and structural statistics closer to the target distribution.

**The Kob-Andersen (KA) system with** $N = 44$ **and 3 species.**   We consider a ternary variant of the Kob–Andersen mixture Kob & Andersen (1995) in two dimensions, with composition $(5/11, 3/11, 3/11)$ (Jung et al., 2023). This system extends the classical binary Lennard–Jones mixture by introducing a third component, which both enhances glass-forming ability (hindering crystallization) and improves the efficiency of the swap Monte Carlo algorithm Parmar et al. (2020). The interaction potential between two particles of species $s_1$ and $s_2$ at nearest-image distance $r$ is

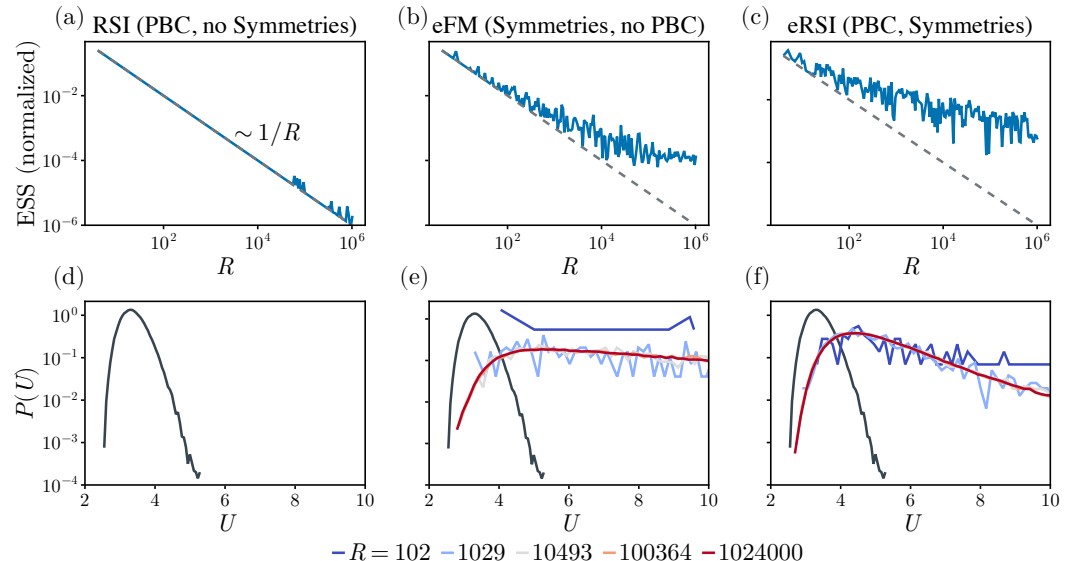

Figure 8: **Deep dive into the effective sample size (ESS) for the $N = 10$ system.** *(Top)*: Normalized ESS (see Appendix E) as a function of the number of samples $R$. This metric reflects the proportion of samples effectively contributing to expectation estimates and provides an upper bound on the estimator variance (Agapiou et al., 2017). *(Bottom)*: Histograms of the target energy observable $U_\star$ for samples generated by each model. Each column corresponds to one model.

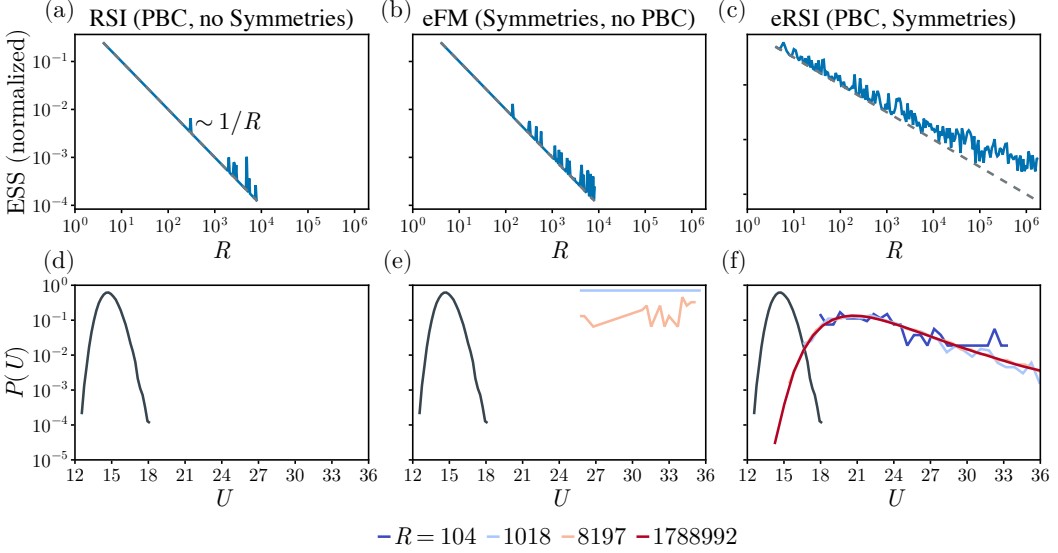

Figure 9: **Deep dive into the effective sample size (ESS) for the $N = 44$ system.** *(Top)*: Normalized ESS (see Appendix E) as a function of the number of samples $R$. This metric reflects the proportion of samples effectively contributing to expectation estimates and provides an upper bound on the estimator variance (Agapiou et al., 2017). *(Bottom)*: Histograms of the target energy observable $U_\star$ for samples generated by each model. Each column corresponds to one model.

defined as

$$
W_{\mathrm{KA}}(s_1, s_2, r) = \begin{cases} 4\epsilon_{s_1 s_2} W_{\mathrm{LJ}}(s_1, s_2, r) + W_0 + W_2 \left(\frac{r}{\sigma_{s_1 s_2}}\right)^2 + W_4 \left(\frac{r}{\sigma_{s_1 s_2}}\right)^4 , & r < 2\sigma_{s_1 s_2}, \\ 0, & \text{otherwise}, \end{cases}
$$

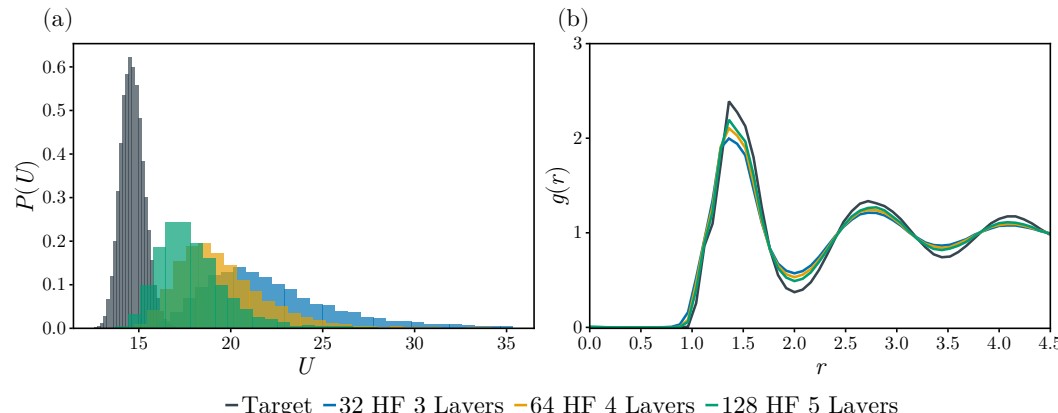

Figure 10: **Comparison of model sizes on the IPL system with** $N = 44$. Variants of the EGNN architecture (14) are evaluated with $K \in 3, 4, 5$ and hidden feature size (i.e., the width of each neural network denoted HF) $\in 32, 64, 128$. All models were trained and selected following the procedure in Appendix F.3. *(Left)*: distribution of energies from 8192 generated samples per model. *(Right)*: corresponding radial distribution functions. Larger architectures yield improved fidelity to the target distribution. All experiments in Section 5 use the smallest network configuration.

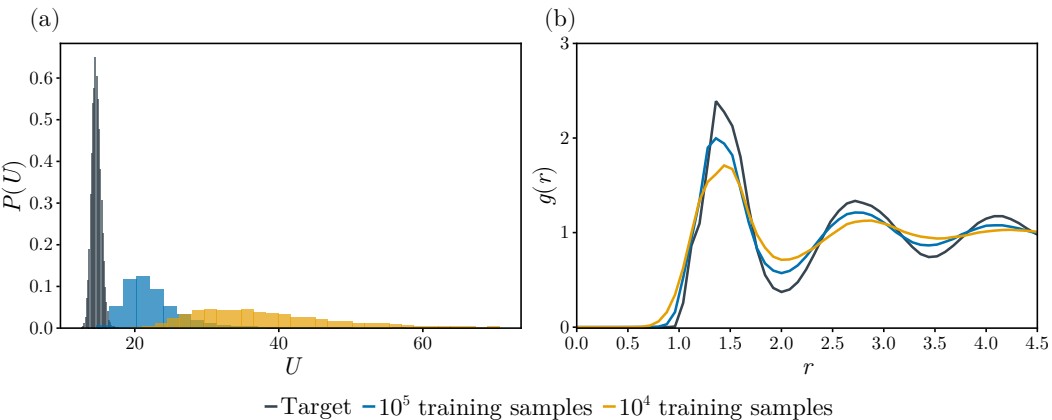

Figure 11: **Comparison of dataset sizes on the IPL system with** $N = 44$. Models were trained on datasets of size $10^4$ and $10^5$, both generated following the procedure described in Appendix F.1. The GNN used has the same architecture as in Appendix F.2. *(Left)*: Energy distribution of 8192 generated samples. *(Right)*: corresponding radial distribution functions. Larger datasets yield notably improved fidelity in both metrics.

with

$$\mathrm{W}_{\mathrm{LJ}}(s_1, s_2, r) = \left[ \left( \frac{\sigma_{s_1 s_2}}{r} \right)^{12} - \left( \frac{\sigma_{s_1 s_2}}{r} \right)^6 \right], \quad \epsilon = \begin{pmatrix} 1.0 & 1.5 & 0.75 \\ 1.5 & 0.5 & 1.5 \\ 0.75 & 1.5 & 0.75 \end{pmatrix}, \quad \sigma = \begin{pmatrix} 1.0 & 0.8 & 0.9 \\ 0.8 & 0.88 & 0.8 \\ 0.9 & 0.8 & 0.94 \end{pmatrix},$$

and correction terms $(\mathrm{W}_0, \mathrm{W}_2, \mathrm{W}_4)$ chosen as in Jung et al. (2023). The particle density is fixed at $N/L^2 = 1.192075$, and we consider two temperatures, $T = 1.0$ and $T = 0.32$.

Datasets were generated with the Metropolis–Hastings Monte Carlo algorithm as in Appendix F.1. At the lowest temperature $T = 0.32$, we augmented the Gaussian displacement moves with "swap" moves (Ninarello et al., 2017), applied with probability $p_{\mathrm{swap}}$. In a swap move proposal, two particles of different species are selected at random and their species are exchanged. The proposal is then accepted or rejected according to the usual Metropolis criterion. Swap moves are known to dramatically accelerate equilibration in this specific system (Jung et al., 2023).

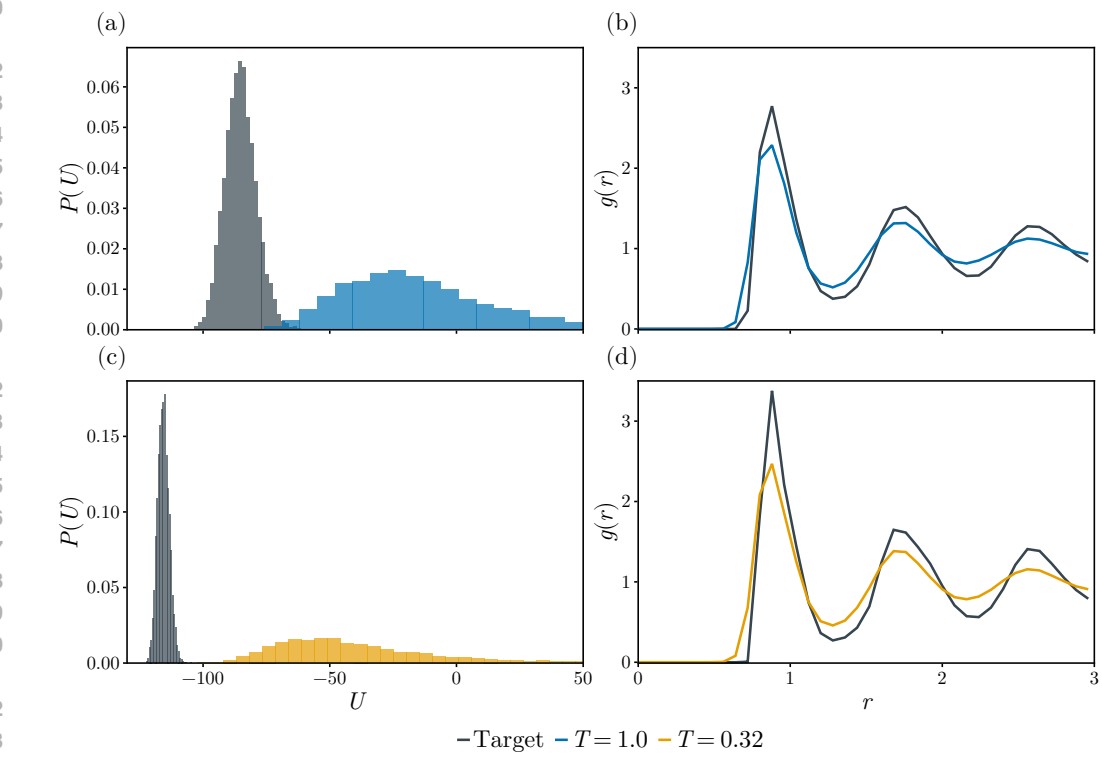

Figure 12: **Energy and structural statistics for the KA system with** $N = 44$ **at two temperatures.** The *left* panels show histograms of the potential energy $U_\star$ evaluated on 8192 samples from the model, while the *right* panels display the corresponding radial distribution functions. Results are shown at $T = 1.0$ *(top row)* and $T = 0.32$ *(bottom row)*.

We trained a simple eRSI model using the smallest GNN architecture (see Appendix F.2), with learning rate $10^{-5}$ and gradient clipping, following the procedure described in Appendix F.3. Figure 12 reports the resulting observables at two temperatures, demonstrating that eRSI produces physically plausible samples. Additional configurations are displayed in Figure 13.

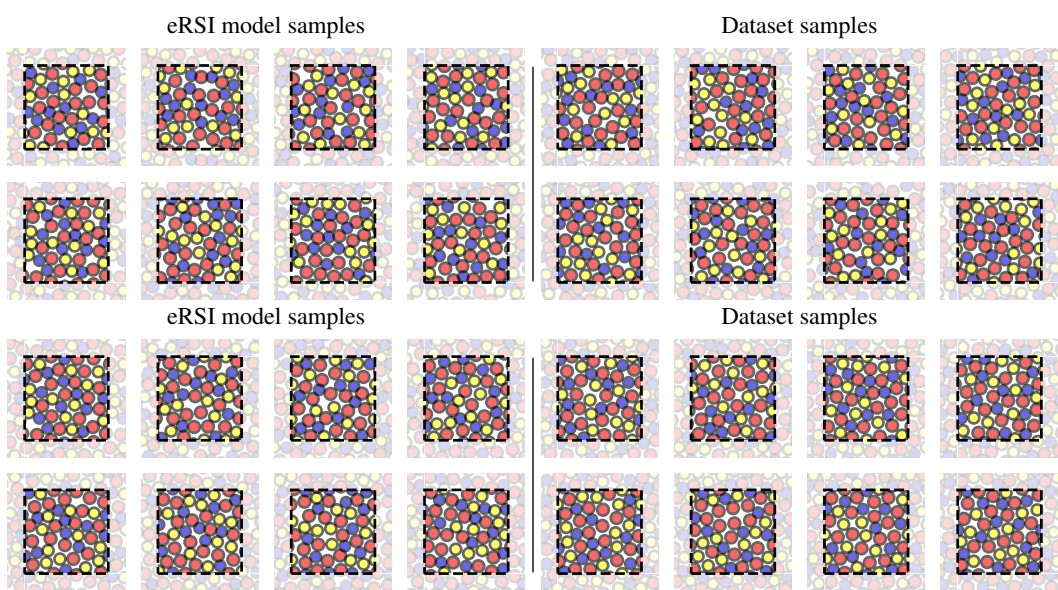

Figure 13: **Additional equilibrium samples for the KA system with** $N = 44$ **at two temperatures.** Configurations generated by the eRSI model *(left)* and reference dataset samples *(right)*, shown at $T = 1.0$ *(top row)* and $T = 0.32$ *(bottom row)*.

