# OpenReview forum: "Riemannian Stochastic Interpolants for Amorphous Particle Systems"
_ICLR.cc/2026/Conference — Submitted to ICLR 2026_

### Official Review · Reviewer_NrV2 · 2025-10-28

**Soundness:** 3
**Presentation:** 3
**Contribution:** 1
**Rating:** 2
**Confidence:** 4

**Summary:**

The paper presents a generative model for amorphous materials based on stochastic interpolants, taking symmetries and periodic boundary conditions into account. Mathematical details regarding the model design are included in detail. The method is evaluated on a small 2-d toy example simulating a single system at two scales (10 and 44 atoms respectively).

**Strengths:**

The paper is well written and while the mathematical details at time makes the paper a bit dense, it is quite readable.
The topic is timely and of great interest.
The method is described in sufficient detail, which makes it likely that results can be reproduced.
The focus on distributional/ensemble metrics is relevant and interesting.

**Weaknesses:**

The experimental validation is very limited. There is no real data example, e.g. with a 3-dimensional amorphous material.
While the method could be expected to generalize between different systems, the experiments only show a model trained for a single system.
The ability of the model to generalize should be clearly demonstrated.

**Questions:**

Could you write M = R^3 / (LZ)^3 ?

Are you sure that Eq. (1) is not a metric on the quotient space M? Perhaps you meant to say it is not a true metric on R^d.

The potential energy as defined in Eq. (2) assumes no self-interaction across the periodic boundaries, right?

What do you mean by "For instance, the modulo operator defines a fundamental invariance for probability densities on M, as any density that is not modulo-invariant would assign infinite mass (...)"? Do you mean p(X) = p(X+kL) for all k in Z^d? I am not sure I understand this, since if X is in M, then X (in R^d) and X+kL (in R^d) is literally the same point in M, so this is already built into the domain, p: M->R.

Line 131. Is the ⨂ notation necessary here? Could you not just write (X + u) % L? I think that is fairly standard and perhaps more readable. Same goes for line 139. Perhaps you prefer the chosen notation because it makes it explicit that operations are on all N atoms?

What is the practical significance of Eq. (4)? Also, if b should be in R^{N|S|+Nd} that seems to not align with the definition of b in appendix A, lemma 10: b=1_N ⨂ c where c is in R^d.

Could the logarithmic map (Definition 2) be written more compactly as (A-B+L/2) % L - L/2 ?

On the flat torus, where the differential of the exponential map is the identity, doesn’t Eq. (9) recover the exact trajectories rather than only the time-marginals?

I am confused about the notation in Proposition 8. The velocity field is defined as \hat v: [0,1] x C -> TC. So is are the coordinates not already defined on M, and thereby by definition \hat v(t, (s,X)) = \hat v(t, (s,X+kL)) ?

What is the relation to generative models based on kinetic Langevin diffusion? Could you compare to this line of work?

Can you demonstrate the capability of the model to generalize beyond the training data?

---

> ### Author Response · Authors · 2025-11-26
> **Answer to reviewer NrV2 [1/3]**
>
> We thank Reviewer **NrV2** for the helpful feedback. Our detailed responses are provided below.
>
> > Could you write M = R^3 / (LZ)^3 ? [...] Are you sure that Eq. (1) is not a metric on the quotient space M? Perhaps you meant to say it is not a true metric on R^d.
>
> Yes, the set $\mathcal{M}$ could indeed be written as $\mathbb{R}^d / (L\mathbb{Z})^d$. In the current manuscript, however, we implicitly allow multiple representatives of the same equivalence class (i.e., we work with $\mathcal{M} = \mathbb{R}^d$) while enforcing periodicity through the metric and invariances. We will clarify this choice and make the notation consistent in the next revision. Regarding Eq. (1), you are correct, we will clarify this in the revision. The function defined in Eq. (1) is a valid pre-metric within the equivalence class of $\mathcal{M}$ (it is non-negative, positive and non-degenerate, which are the minimal requirements needed for Riemannian Flow Matching (see Section 3.2 of [1])) and it is also a valid distance as it satisfies the triangular inequality. However, as you point out, the same function is not a true distance on $\mathbb{R}^d$, since points differing by a lattice vector have zero distance. We will adjust the wording accordingly.
>
> > Line 131. Is the $\otimes$ notation necessary here? Could you not just write (X + u) % L? I think that is fairly standard and perhaps more readable. Same goes for line 139. Perhaps you prefer the chosen notation because it makes it explicit that operations are on all N atoms?
>
> The reviewer’s interpretation is correct: we used the tensor-product notation to make explicit that, in the translation, the same $d$-dimensional vector is added to every particle, and in the symmetry, the same $d$-dimensional matrix is applied to all particles.
>
> > Could the logarithmic map (Definition 2) be written more compactly as (A-B+L/2) % L - L/2 ? [...] On the flat torus, where the differential of the exponential map is the identity, doesn’t Eq. (9) recover the exact trajectories rather than only the time-marginals?
>
> For the logarithmic map, we rely on standard expressions for the flat torus (summarized, for example, in Table 5 of [1]). The compact expression you suggest corresponds exactly to our Proposition 5, and we will highlight this more clearly in the revised version. Regarding Eq. (9): even in the Euclidean case, the interpolant $X_t = \exp_{X_0}(t\log_{X_0}(X_1)) = (1-t)X_0 + tX_1$ is non-Markovian by construction, since it depends simultaneously on the endpoints $(X_0, X_1)$. By contrast, the solution of the ODE in Eq. (5) is always Markovian. Therefore, the trajectory of the ODE can never coincide exactly with the interpolant trajectory. Even in Euclidean space, optimizing Eq. (9) recovers the velocity field whose associated Markov process matches the interpolant only at the time marginals (it is a Markovian projection of the non-Markovian interpolant).
>
> > The potential energy as defined in Eq. (2) assumes no self-interaction across the periodic boundaries, right?
>
> In Eq. (2), each particle interacts with all other particles using their minimal periodic (modulo-$L$) representation as defined in Eq. (1). Because we always take the minimal-image distance, no particle ever interacts with one of its own periodic replicas. Thus, there are indeed no self-interactions across the periodic boundaries.
>
> > What is the practical significance of Eq. (4)? Also, if b should be in R^{N|S|+Nd} that seems to not align with the definition of b in appendix A, lemma 10: b=1_N $\otimes$ c where c is in R^d.
>
> Eq. (4)  shows that the group actions can be expressed as a composition of a projection with an underlying affine action. This affine structure is what allows us to define velocity-field equivariance in Definition 7. Regarding the vector $b$, you are absolutely right,we will correct this in the revision.
>
> > What do you mean by "For instance, the modulo operator defines a fundamental invariance for probability densities on M, as any density that is not modulo-invariant would assign infinite mass (...)"? Do you mean p(X) = p(X+kL) for all k in Z^d? I am not sure I understand this, since if X is in M, then X (in R^d) and X+kL (in R^d) is literally the same point in M, so this is already built into the domain, p: M->R. [...] I am confused about the notation in Proposition 8. The velocity field is defined as \hat v: [0,1] x C -> TC. So is are the coordinates not already defined on M, and thereby by definition \hat v(t, (s,X)) = \hat v(t, (s,X+kL)) ?
>
> Both questions point to the same underlying issue. If we view these objects as functions on $\mathbb{R}^d$, then they must be invariant under translations by multiples of $L$. In the current manuscript, however, $\mathcal{M}$ still contains multiple representatives of the same point on the torus, which creates notational ambiguity regarding when such invariance is enforced. We will update the manuscript accordingly to make this explicit and avoid confusion.

---

> > ### Author Response · Authors · 2025-11-26
> > **Answer to reviewer NrV2 [2/3]**
> >
> > > What is the relation to generative models based on kinetic Langevin diffusion? Could you compare to this line of work?
> >
> > To the best of our knowledge, there is no direct connection between our approach and generative models based on kinetic Langevin diffusion. Kinetic Langevin dynamics constitute a different class of generative models, and while they have been successfully used in related settings (for example, in [2] for modeling crystal structures) they do not appear to have a specific conceptual or methodological link to our work.
> >
> > > The experimental validation is very limited. There is no real data example, e.g. with a 3-dimensional amorphous material [...] the experiments only show a model trained for a single system.
> >
> > The realism of our work is comparable to the current state-of-the-art, as the dimensionality of our investigated systems, such as the inverse power law and Kob-Andersen (App. F4) potentials, matches that of standard Lennard-Jones systems studied in related generative modeling literature [3,4,5,6]. From the viewpoint of modelling amorphous solids, 3D models are not more real than 2D ones, as experimental realisations exist for both, and both dimensions are known to share essentially the same physics. Crucially, our models address greater complexity by being defined on non-Euclidean spaces and involving multiple species, making it notoriously challenging to sample [10]. Furthermore, our objective is fundamentally distinct from recent works on crystals [7,8,9] (which also feature non-euclidean representations and multiple species), which focus on reproducing a dataset; instead, our approach is centered on  sampling by leveraging the density, a capability validated by our re-weighted results. More details about this aspect are given in the global answer to referees.
> >
> > > The ability of the model to generalize should be clearly demonstrated. [...] While the method could be expected to generalize between different systems, the experiments only show a model trained for a single system. [...] Can you demonstrate the capability of the model to generalize beyond the training data?
> >
> > The reviewer's point on generalization is highly relevant. While our current focus is on the challenging task of sampling a single, complex system, we could expect our model to generalize. Our methodology is closely related to Boltzmann Generators [10], which have been recently shown to be transferable across different particle systems [12]. Given this, demonstrating the model's transferability to new systems is a clear and important objective we designate for future work.

---

> > > ### Author Response · Authors · 2025-11-26
> > > **Answer to reviewer NrV2 [3/3]**
> > >
> > > **References**
> > >
> > > [1] Ricky T. Q. Chen, & Yaron Lipman (2024). Flow Matching on General Geometries. In The Twelfth International Conference on Learning Representations.
> > >
> > > [2] Cornet, F., Bergamin, F., Bhowmik, A., Garcia-Lastra, J., Frellsen, J., & Schmidt, M. (2025). Kinetic Langevin Diffusion for Crystalline Materials Generation. In Proceedings of the 42nd International Conference on Machine Learning (pp. 11215–11244). PMLR.
> > >
> > > [3] Jonas Köhler, Leon Klein, & Frank Noé. (2019). Equivariant Flows: sampling configurations for multi-body systems with symmetric energies.
> > >
> > > [4] Kohler, J., Klein, L., & Noe, F. (2020). Equivariant Flows: Exact Likelihood Generative Learning for Symmetric Densities. In Proceedings of the 37th International Conference on Machine Learning (pp. 5361–5370). PMLR.
> > >
> > > [5] Wirnsberger, P., Papamakarios, G., Ibarz, B., Racanière, S., Ballard, A., Pritzel, A., & Blundell, C. (2022). Normalizing flows for atomic solids. Machine Learning: Science and Technology, 3(2), 025009.
> > >
> > > [5] Klein, L., Krämer, A., & Noe, F. (2023). Equivariant flow matching. In Advances in Neural Information Processing Systems (pp. 59886–59910). Curran Associates, Inc..
> > >
> > > [6] Midgley, L., Stimper, V., Antorán, J., Mathieu, E., Schölkopf, B., & Hernandez-Lobato, J. (2023). SE(3) Equivariant Augmented Coupling Flows. In Advances in Neural Information Processing Systems (pp. 79200–79225). Curran Associates, Inc..
> > >
> > > [7] Sherry Yang, KwangHwan Cho, Amil Merchant, Pieter Abbeel, Dale Schuurmans, Igor Mordatch, & Ekin Cubuk (2023). Scalable Diffusion for Materials Generation. In NeurIPS 2023 AI for Science Workshop.
> > >
> > > [8] Rui Jiao, Wenbing Huang, Peĳia Lin, Jiaqi Han, Pin Chen, Yutong Lu, & Yang Liu (2023). Crystal Structure Prediction by Joint Equivariant Diffusion. In Thirty-seventh Conference on Neural Information Processing Systems.
> > >
> > > [9] Miller, B., Chen, R., Sriram, A., & Wood, B. (2024). FlowMM: Generating Materials with Riemannian Flow Matching. In Proceedings of the 41st International Conference on Machine Learning (pp. 35664–35686). PMLR.
> > >
> > > [10] Berthier, L., & Reichman, D. (2023). Modern computational studies of the glass transition. Nature Reviews Physics, 5(2), 102–116.
> > >
> > > [11] Frank Noé, Simon Olsson, Jonas Köhler, & Hao Wu (2019). Boltzmann generators: Sampling equilibrium states of many-body systems with deep learning. Science, 365(6457), eaaw1147.
> > >
> > > [12] Klein, L., & Noé, F. (2024). Transferable Boltzmann Generators. In Advances in Neural Information Processing Systems (pp. 45281–45314). Curran Associates, Inc..

---

> ### Comment · Reviewer_NrV2 · 2025-11-26
>
> >>Could you write M = R^3 / (LZ)^3 ? [...] Are you sure that Eq. (1) is not a metric on the quotient space M? Perhaps you meant to say it is not a true metric on R^d.
>
> >Yes...
>
> Resolved
>
> >>Line 131. Is the notation necessary here?
>
> >... we used the tensor-product notation...
>
> Resolved
>
> >>Could the logarithmic map (Definition 2) be written more compactly
>
> > ... we rely on standard expressions ...
>
> I prefer the more compact expression, as it is simpler, more intuitive, and is how you would compute it in practice.
>
> >>On the flat torus, where the differential of the exponential map is the identity, doesn’t Eq. (9) recover the exact trajectories rather than only the time-marginals?
>
> >...optimizing Eq. (9) recovers the velocity field whose associated Markov process matches the interpolant only at the time marginals...
>
> Got it - it recovers the exact sampling path for some specific x1 (i.e only when conditioned on x1). Resolved
>
>
> >>The potential energy as defined in Eq. (2) assumes no self-interaction across the periodic boundaries, right?
>
> >... no particle ever interacts with one of its own periodic replicas...
>
> Resolved (maybe it would be nice to write this explicitly.)
>
> >>What is the practical significance of Eq. (4)?
>
> >Eq. (4) shows that the group actions can be expressed as a composition of a projection with an underlying affine action. This affine structure is what allows us to define velocity-field equivariance in Definition 7.
>
> >>Also, if b should ... not align with the definition of b in appendix A...
>
> >Regarding the vector, you are absolutely right,we will correct this in the revision.
>
> Resolved
>
> >>What do you mean by "For instance, the modulo operator (...) I am confused about the notation in Proposition 8. ...
>
> >... notational ambiguity (...) update the manuscript accordingly to make this explicit and avoid confusion.
>
> Resolved
>
> >>What is the relation to generative models based on kinetic Langevin diffusion? Could you compare to this line of work?
>
> >... there is no direct connection...
>
> Okay - I would have thought there would be a rich connection to this literature.
>
> >>The experimental validation is very limited. There is no real data example, e.g. with a 3-dimensional amorphous material [...] the experiments only show a model trained for a single system.
>
> >...The realism of our work is comparable to the current state-of-the-art...
>
> >>The ability of the model to generalize should be clearly demonstrated.
>
> >...demonstrating the model's transferability to new systems is a clear and important objective we designate for future work.
>
> I still find that the limited evaluation and absence of demonstrated generalization remains a major shortcoming.

---

> > ### Author Response · Authors · 2025-11-26
> >
> > Thank you again for carefully reading our responses. We would like to briefly clarify two remaining points raised in your latest comment.
> >
> > First, regarding the claim that the experimental validation is "limited": as you also acknowledged in your summary of our response, our evaluation is in fact fully aligned with the current state of the art. Specifically, (i) we focus on systems of the same dimensional scale as the strongest recent baselines; (ii) we target settings that introduce new and harder challenges (non-Euclidean representations, additional invariances, ...); and (iii) these systems are more aligned with the physics literature than those used in prior works such as [4,5,6]. For these reasons, we believe our evaluation is not only adequate but more demanding than what is typically used in this line of research.
> >
> > Second, regarding "generalization", if the reviewer refers to transferring a learned model across different physical systems, this remains essentially unexplored in the literature. The closely related works [4,5,6] do not address it, and only very recent papers such as [12] focus exclusively on this question without tackling any of the other challenges we consider. Given this context and the already substantial technical contributions of our paper, we view cross-system generalization as outside our scope. If the reviewer had a different notion of generalization in mind, we would be grateful for clarification.

---

### Official Review · Reviewer_kVyB · 2025-10-29

**Soundness:** 3
**Presentation:** 2
**Contribution:** 3
**Rating:** 6
**Confidence:** 2

**Summary:**

The paper proposes to use Riemannian stochastic interpolants frameowork and group equivariant network for amorphous systems generation. The authors also adapt the architecture of graph neural network to leverage the full symmetry of the amorphous materials. Experiments on a classical glass model show the empirical performance of the proposed methods.

**Strengths:**

1. The paper is well-structured and esay to follow.

2. The paper leverages the symmetry and geometry structure of the amorphous materials, which is reasonable,

**Weaknesses:**

1. Limited experimental scope. The experiments appear to be restricted to two-dimensional and relatively small-scale datasets. Additional experiments on larger-scale and real-world datasets would strengthen the paper’s contributions. Please refer to Question 1 for further details.

2. Lack of efficiency analysis. I noticed that the authors use Eq. (8) to compute the expectation of physical quantities. To the best of my knowledge, such likelihood computations can be inefficient and inaccurate. Additional experimental results are needed to demonstrate that the proposed estimation process indeed converges reliably.

**Questions:**

1. Since I am not an expert in amorphous materials, could you clarify whether there are any real-world datasets or tasks in this domain that are suitable for diffusion models? As mentioned in Weakness 1, experiments on large-scale datasets would help demonstrate the scalability of the proposed method and further strengthen the contribution of this work.

2. Could you elaborate on how you evaluate the average potential energy and heat? As mentioned in Weakness 2, does the simulation of Eq. (8) converge in practice?

3. How does the choice of numerical ODE solver affect the generation performance?

---

> ### Author Response · Authors · 2025-11-26
> **Answer to reviewer kVyB [1/2]**
>
> We are grateful to Reviewer **kVyB** for their careful assessment of our work. We respond to the reviewer’s questions and comments below.
>
> > The experiments appear to be restricted to two-dimensional and relatively small-scale datasets. Additional experiments on larger-scale and real-world datasets would strengthen the paper’s contributions. [...] Could you clarify whether there are any real-world datasets or tasks in this domain that are suitable for diffusion models? [...] Experiments on large-scale datasets would help demonstrate the scalability of the proposed method and further strengthen the contribution of this work.
>
> The global response also addresses this point. Unlike typical generative modeling tasks aimed at reproducing experimental datasets, our work focuses on sampling from a known Boltzmann distribution (Eq. 3) defined by a given potential function U. Like us, prior works have considered potentials arising in particle systems [1,2,3,4,5], with comparable system dimensionality (number of particles × spatial dimensions + species). Importantly, our models introduce additional complexity (particularly compared to [1,2,4,5]) : they operate on non-Euclidean spaces and involve multiple species, making sampling substantially more challenging. Although the inverse-power-law and Kob-Andersen potentials (App. F.4) may appear simple, they are canonical models of metallic glass formers that exhibit the full phenomenology of glass physics, even for small particle counts [6,7], making it notoriously difficult to sample
>
> As noted in Weakness 2, computing likelihoods in these high-dimensional systems is computationally expensive and scales poorly. Nevertheless, likelihood evaluation is essential to our method, enabling unbiased reweighting and accurate computation of physical observables. While this imposes practical limits on scalability, it is worth noting that classic learning-free MCMC methods typically struggle to reach equilibration in these systems, which is precisely why we adopt a reweighting-based approach instead. We will make this limitation more explicit in the next revision.
>
> > I noticed that the authors use Eq. (8) to compute the expectation of physical quantities. To the best of my knowledge, such likelihood computations can be inefficient and inaccurate. [...] How does the choice of numerical ODE solver affect the generation performance?
>
> We thank the reviewer for this question. The central motivation of our work is to leverage a generative model with tractable likelihoods to compute expectations of physical quantities, making Eq. (8) a key component. We agree with the reviewer that Eq. (8) has practical limitations, as its divergence term makes simulation costly; we will make this limitation more explicit in the revised manuscript. In our implementation, we use the dopri5 adaptive solver from the torchdiffeq library for both Eq. (5) and Eq. (8), which evaluates the velocity field approximately 150 times per full trajectory. Comparisons with long Euler simulations (over a thousand steps) show no significant deviations, confirming the accuracy of our log-likelihood computation. An additional validation comes from our ability to reproduce the observables relevant to physics, by order of difficulty: mean energy $U$, specific heat $c_V$ and radial distribution $g(r)$ (Fig. 3 (a)-(b)).
>
> > Additional experimental results are needed to demonstrate that the proposed estimation process indeed converges reliably. Could you elaborate on how you evaluate the average potential energy and heat?
>
> In our setting, the potential function $\mathrm{U}_{\star}$ is known explicitly, so it can be evaluated at any point. Additionally, we have access to true samples generated using gold-standard samplers (see App. F), which, while not always applicable, are valid for the systems studied here. These samples are used to compute reference values (labeled “Target” in the figures) for the average potential energy and heat. For the model, values are obtained in two ways: (i) by generating samples via simulation of Eq. (5) (labeled “Model” in the figures), or (ii) by computing expectations using Importance Sampling with Eq. (8), as detailed in App. E (labeled “Reweight” in the figures). We will clarify this procedure further in the next revision.
>
> > Does the simulation of Eq. (8) converge in practice?
>
> As recalled above, in Fig. 3 (a)-(b), we observe that the computed expectations of physically relevant observables converge reliably to these known targets, confirming that the model faithfully approximates the Boltzmann distribution. In other words, the inherent difficulty of computing Eq. (8) does not prevent rigorous validation.

---

> > ### Author Response · Authors · 2025-11-26
> > **Answer to reviewer kVyB [2/2]**
> >
> > **References**
> >
> > [1] Jonas Köhler, Leon Klein, & Frank Noé. (2019). Equivariant Flows: sampling configurations for multi-body systems with symmetric energies.
> >
> > [2] Kohler, J., Klein, L., & Noe, F. (2020). Equivariant Flows: Exact Likelihood Generative Learning for Symmetric Densities. In Proceedings of the 37th International Conference on Machine Learning (pp. 5361–5370). PMLR.
> >
> > [3] Wirnsberger, P., Papamakarios, G., Ibarz, B., Racanière, S., Ballard, A., Pritzel, A., & Blundell, C. (2022). Normalizing flows for atomic solids. Machine Learning: Science and Technology, 3(2), 025009.
> >
> > [4] Klein, L., Krämer, A., & Noe, F. (2023). Equivariant flow matching. In Advances in Neural Information Processing Systems (pp. 59886–59910). Curran Associates, Inc..
> >
> > [5] Midgley, L., Stimper, V., Antorán, J., Mathieu, E., Schölkopf, B., & Hernandez-Lobato, J. (2023). SE(3) Equivariant Augmented Coupling Flows. In Advances in Neural Information Processing Systems (pp. 79200–79225). Curran Associates, Inc..
> >
> > [6] Berthier, L. & Biroli, G. (2011).  Theoretical perspective on the glass transition and amorphous materials. Reviews of Modern Physics, 83(2):587–645
> >
> > [7] Berthier, L., & Reichman, D. (2023). Modern computational studies of the glass transition. Nature Reviews Physics, 5(2), 102–116

---

### Official Review · Reviewer_73q7 · 2025-10-30

**Soundness:** 3
**Presentation:** 2
**Contribution:** 1
**Rating:** 2
**Confidence:** 3

**Summary:**

The paper introduces an equivariant form of stochastic interpolants and applies it to amorphous materials. The main results are given on a toy problem with either 11 or 44 atoms with two different species.

**Strengths:**

- Figure 2 makes the equivariances quite convincing
- Competing methods have drawbacks (Riemannian DDPM, no likelihood; maximum-likelihood training of ODE, expensive)
- The reweighing is possible via the framework and it is shown to make a big improvement in performance.

**Weaknesses:**

- Figure 1, right side. The word "symmetrized" is written next to an image in which the symmetry is far from clear. What's going on there?
- There is a lot of time spent on what I would consider background information. Many of these symmetries are closely discussed in other works. I agree that a specific treatment of stochastic interpolants is technically new, but the details discussed here are somewhat minor.
- While the performance is obviously better with the author's treatment, the results are rather simple. Of course, everything gets more complex in amorphous state with periodic boundary conditions, but 44 atom with two species is not very many or complex. Is there any experimental data you can attempt to fit to?

**Questions:**

- What is the citation issue on line 177?
- Can you consider any bigger systems or ones with more motivated datasets?
- Can you explain your main contribution more clearly? It seems like many of these modeling aspects existed already in extremely related frameworks.

---

> ### Author Response · Authors · 2025-11-26
> **Answer to reviewer 73q7 [1/2]**
>
> We thank Reviewer **73q7** for their thoughtful feedback. Below, we address each of the reviewer’s concerns in detail
>
> > There is a lot of time spent on what I would consider background information. Many of these symmetries are closely discussed in other works. I agree that a specific treatment of stochastic interpolants is technically new, but the details discussed here are somewhat minor. [...] Can you explain your main contribution more clearly? It seems like many of these modeling aspects existed already in extremely related frameworks.
>
> We thank the reviewer for this comment and will improve the clarity of our exposition. Our main contribution is a rigorous incorporation of equivariances in the Riemannian Stochastic Interpolant (RSI) framework, so that it can be applied to sampling multi-species particle systems with periodic boundary conditions, for non-periodic configurations. Specifically:
>
> 1. **Incorporating invariances into RSI.** We extend the intuitive framework of equivariant Continuous Normalizing Flows (eCNFs) [4] to non-Euclidean manifolds with nonlinear invariances. In addition, Propositions 4 and 9 are specific to RSI and establish precise conditions under which the optimal velocity field is equivariant, results that were not available in equivariant Flow Matching (eFM) [5].
>
> 2. **Facing a challenging sampling problem.** While prior works have explored sampling equilibrium distributions of particle systems using eCNFs or eFM [4,5,6], these efforts primarily focused on Euclidean, single-species settings such as Lennard-Jones systems. Our work instead tackles non-Euclidean, multi-species systems at comparable dimensionality, a shift that fundamentally alters the system representation, geometry, and invariances and is concretely relevant to study amorphous solids. Although some studies have addressed non-Euclidean multi-species systems in crystalline materials [7,8], crystal configurations differfrom amorphous systems as they are represented via lattices uniquely defined by Nigli reductions, leading to different invariance structures and modeling requirements. Moreover, theoretical treatments of invariances in those works are often incomplete (e.g., the ungrounded translation-invariance condition for conditional velocity fields in Eq. 14 of [7]), whereas our approach consistently provides formal invariance guarantees. Finally, while crystal-focused works typically perform generative modeling (sampling from an unknown distribution given data), we perform sampling from a known potential energy, not merely from an observed dataset of examples.
>
> > Of course, everything gets more complex in amorphous state with periodic boundary conditions, but 44 atom with two species is not very many or complex. Is there any experimental data you can attempt to fit to? Can you consider any bigger systems or ones with more motivated datasets?
>
> We thank the reviewer for the opportunity to clarify this point. Our objective is fundamentally different from generative modelling approaches that aim to reproduce an experimental dataset: our goal is to sample the canonical Boltzmann distribution (Eq; 3), which is fully determined by the potential energy function U. The challenge does not lie in reproducing a dataset but in sampling from highly non-trivial energy landscapes. As discussed in the global response, the systems we study (the inverse-power-law and the Kob–Andersen (App. F.4) potentials) are canonical models of metallic glass formers and are extensively used in theoretical and computational studies of amorphous materials [1,2]. Although they may appear simple at first sight, they already display the full phenomenology of glass physics: extremely slow and heterogeneous dynamics, strong resistance to crystallisation, and rich mechanical behaviour, even at small particle counts. These models are also notoriously challenging to simulate at low temperature, even for modest system sizes [3]. In fact, we disagree with the statement that the systems we consider are not “complex”: they are extremely hard to sample. We also point out that prior work on sampling particle systems using closely related methodologies [4,5,6] operates at comparable dimensionalities, despite targeting considerably simpler settings (e.g., single species, standard Euclidean geometry, and no glassy dynamics). We will revise the manuscript to clarify these points and better motivate the chosen systems.

---

> > ### Author Response · Authors · 2025-11-26
> > **Answer to reviewer 73q7 [2/2]**
> >
> > > Figure 1, right side. The word "symmetrized" is written next to an image in which the symmetry is far from clear. What's going on there?
> >
> > We acknowledge that this representation may appear ambiguous. The illustrated transformation represents a counterclockwise 90° rotation, i.e., axial symmetry with respect to the line going from the bottom left corner to the top right corner. This is one of the possible actions of the hyperoctahedral group, which is the symmetry group of the hypercube. We will update the figure and caption to make this interpretation explicit.
> >
> > > What is the citation issue on line 177?
> >
> > We inadvertently omitted the publication year. We will correct this in the revised manuscript.
> >
> >
> > **References**
> >
> > [1] Berthier, L. & Biroli, G. (2011).  Theoretical perspective on the glass transition and amorphous materials. Reviews of Modern Physics, 83(2):587–645
> >
> > [2] Berthier, L., & Reichman, D. (2023). Modern computational studies of the glass transition. Nature Reviews Physics, 5(2), 102–116.
> >
> > [3] Jung, G., Ozawa, M., Biroli, G., Berthier, L. (2025). Numerical investigation of the equilibrium Kauzmann transition in a two-dimensional atomistic glass. arXiv:2507.03590.
> >
> > [4] Kohler, J., Klein, L., & Noe, F. (2020). Equivariant Flows: Exact Likelihood Generative Learning for Symmetric Densities. In Proceedings of the 37th International Conference on Machine Learning (pp. 5361–5370). PMLR.
> >
> > [5] Klein, L., Krämer, A., & Noe, F. (2023). Equivariant flow matching. In Advances in Neural Information Processing Systems (pp. 59886–59910). Curran Associates, Inc..
> >
> > [6] Midgley, L., Stimper, V., Antorán, J., Mathieu, E., Schölkopf, B., & Hernandez-Lobato, J. (2023). SE(3) Equivariant Augmented Coupling Flows. In Advances in Neural Information Processing Systems (pp. 79200–79225). Curran Associates, Inc..
> >
> > [7] Miller, B., Chen, R., Sriram, A., & Wood, B. (2024). FlowMM: Generating Materials with Riemannian Flow Matching. In Proceedings of the 41st International Conference on Machine Learning (pp. 35664–35686). PMLR.
> >
> > [8] Sriram, A., Miller, B., Chen, R., & Wood, B. (2024). FlowLLM: Flow Matching for Material Generation with Large Language Models as Base Distributions. In Advances in Neural Information Processing Systems (pp. 46025–46046). Curran Associates, Inc.

---

> ### Comment · Reviewer_73q7 · 2025-11-26
> **thanks for the clear response**
>
> I will reply more fully in a bit, but I want to ask something quickly to understand better and give you enough time to respond.
>
> Your method learns to sample? Can you please clearly identify the loss function you are optimizing and how you are sampling? Even when I look back, it is not 100% obvious to me now. The equation (9) looks like a data-based loss. If you don't have samples $X_1$ how are you drawing the paths? It looks like $X_1$ is drawn from $p_\ast$, but I don't see where you specify how to draw from that distribution. If you use another method (mcmc, rejection, etc), then I would classify your contribution as a generative model, not a model learning to sample.
>
> When I read the paper I thought this was a generative model fit to data. If you can point these things out, then I would have a significantly different reading of the complexity of the problem

---

> ### Author Response · Authors · 2025-11-27
> **Training adaptively (without samples a priori) is the next step now we have shown reweighting is possible**
>
> Thank you for the quick follow-up and for pointing us to this need for clarification.
>
> Our ultimate goal is indeed to sample from the knowledge of the energy function and therefore not to rely on previously obtained samples to train the stochastic interpolant (SI) model. A series of recent works focuses on how to train diffusion models (closely related to SI) solely energy-based [a,b,c] but does not consider how to debias/reweight the samples of the learned flow, which is, as we show again here, a crucial step to build trustworthy estimators of physical quantities (here, eFM generates visually plausible samples (Fig. 2) while still yielding poor estimates of expectations ( Fig. 4)).
>
> Here, we take a different route inspired from previous neural samplers relying on normalizing flows and autoregressive models. Showing first that reweighting is possible in experiments where samples were used to train, we will aim next to show that this reweighting allows us to implement the sample-free adaptive methods previously proposed for flows and autoregressive models -- as stated in the final sentence of our “Conclusion and limitations” section:
>
> *Finally, while we used extensive training sets in our experiments, we shall next investigate training methods that alleviate this requirement, such as adaptive MCMCs Gabrié et al. (2022) possibly combined with sequential tempering Wu et al. (2019); McNaughton et al. (2020); Bono et al. (2025) as previously proposed for normalizing flows and autoregressive models.*
>
> We shall clarify this point earlier in the manuscript in the next version.
>
>
> [a] Akhound-Sadegh, Tara, Jarrid Rector-Brooks, Joey Bose, et al. “Iterated Denoising Energy Matching for Sampling from Boltzmann Densities.” Paper presented at Forty-first International Conference on Machine Learning. June 6, 2024. https://openreview.net/forum?id=gVjMwLDFoQ.
>
> [b]Richter, Lorenz, and Julius Berner. “Improved Sampling via Learned Diffusions.” Paper presented at The Twelfth International Conference on Learning Representations. October 13, 2023. https://openreview.net/forum?id=h4pNROsO06.
>
> [c] Havens, Aaron J., Benjamin Kurt Miller, Bing Yan, et al. “Adjoint Sampling: Highly Scalable Diffusion Samplers via Adjoint Matching.” Paper presented at Forty-second International Conference on Machine Learning. June 18, 2025. https://openreview.net/forum?id=6Eg1OrHmg2.

---

### Official Review · Reviewer_WnuX · 2025-10-30

**Soundness:** 3
**Presentation:** 3
**Contribution:** 1
**Rating:** 4
**Confidence:** 2

**Summary:**

The authors tackle the problem of sampling equilibrium configurations of glass-forming materials. They do so by combining the Riemannian Stochastic Interpolants and equivariant flow matching for the groups of interest (permutations, translations and symmetries). They substantiate their claims empirically and improve on baselines where available.

**Strengths:**

- The paper is well-presented.
- The developed theoretical framework is arguably simple, but rigorous and thorough. The proofs seem sound and are well written.
- The paper formalises and proves some intuitive claims (e.g., Prop. 9) – something often overlooked.
- The empirical evidence clearly shows improvement on existing baselines on the provided experiment. Not being an expert in the field of the application, I cannot exactly judge of its quality, however; but the overall method seems to produce much more stable configurations by about a magnitude. Moreover, it looks sound.
- The empirical evidence seems very thorough on the provided dataset.

**Weaknesses:**

- The novelty is arguably very low. This paper mostly applies equivariant architectures to Riemannian Flow Matching. In particular, the considered GNN is made Lipschitz-bounded and equivariant, which, as mentioned in the paper, has already been done numerous times. (Perhaps not all at once?)
- Similarly, the theoretical framework does not seem particularly insightful; it mostly formalises results that intuitively seem self-evident. While it is good to prove these, it does not add anything new. At least, this is in my current understanding of the theorems.
- The experiments are convincing, but arguably most of the improvement comes from the equivariance of the architecture.

**Questions:**

Overall, this seems to be a good paper, but I mostly doubt that ICLR is the right venue: the interesting part for a Machine Learning conference is really the experiments section. Moreover, and because of that, I am not expert on this particular sub-field, so I do apologise in advance for not being knowledgeable enough on this. So:

- Have I missed out on some particular novelty in the paper?
- Are the experiments more related to ML than I have understood? Are there perhaps any conclusions to be made about Riemannian Flow Matching/Stochastic Interpolants?
- Have you tried out your method on different data/larger scales, and do you see similar improvements?
- Could you point out the main differences between your method and Equivariant Flow Matching?

I am happy to engage in the discussion period well to hopefully deepen my understanding and better my assessment of this work.

---

> ### Author Response · Authors · 2025-11-26
> **Answer to reviewer WnuX [1/2]**
>
> We thank Reviewer **WnuX** for their valuable feedback on our work. Below we provide a detailed answer to the reviewer’s concerns.
>
> > I mostly doubt that ICLR is the right venue: the interesting part for a Machine Learning conference is really the experiments section [...] Are the experiments more related to ML than I have understood? Are there perhaps any conclusions to be made about Riemannian Flow Matching/Stochastic Interpolants?
>
> It is not unusual in the ML literature to adapt existing generative modeling frameworks to sample particle systems. Several highly cited examples include: [1] (ICML), which tackles Lennard-Jones potentials by building invariances in Continuous Normalizing Flows; [2] (NeurIPS), which addresses the same potentials by building invariances in Flow Matching; [3] (NeurIPS), which addresses the same potentials by building invariances Normalizing Flows; [4] (NeurIPS), which performs generative modeling on crystals via equivariant Riemannian diffusion; and [5,6] (ICML/NeurIPS), which model crystals using Riemannian Flow Matching.
>
> Our work follows this established line of research: we address potentials arising in amorphous systems by incorporating the appropriate invariances into Riemannian Stochastic Interpolants. This aligns with the broader AI4Science trend of tailoring generative models to physical domains, [a direction increasingly supported at major ML venues](https://github.com/AI4QC/AI_for_Science_paper_collection). Beyond their relevance for physical sciences, particle systems provide configurations whose structure and properties have been thoroughly studied in the last decades in physics. In consequence, they provide a very useful testbed for generative models.
>
> > The novelty is arguably very low. This paper mostly applies equivariant architectures to Riemannian Flow Matching. In particular, the considered GNN is made Lipschitz-bounded and equivariant, which, as mentioned in the paper, has already been done numerous times. [...] Similarly, the theoretical framework does not seem particularly insightful; it mostly formalises results that intuitively seem self-evident. While it is good to prove these, it does not add anything new. At least, this is in my current understanding of the theorems. [...] Have I missed out on some particular novelty in the paper? [...] Could you point out the main differences between your method and Equivariant Flow Matching?
>
> As noted above, similar methodologies do exist, but our contribution differs in several important ways.
>
> 1. **Substantive increase in problem difficulty.** Previous methods [1,2,3] focus on single-species Lennard–Jones systems in Euclidean space. In contrast, we study multi-species systems defined on non-Euclidean manifolds, which introduces several non-trivial challenges: different invariance structures, non-Euclidean geometry, the need for newly designed equivariant architectures, which results in another difficult sampling problem [7].
> 2. **Theoretical novelty.** Prior works [1,2,3] develop intuitive theoretical frameworks for sampling equilibrium particle distributions in Euclidean settings. Our work extends this theory to non-Euclidean manifolds and multi-species systems, and we provide new results showing that the optimal models preserve the required invariances (Props. 4 and 9). These extensions are nontrivial: the geometry, the invariance structure, and the analytical tools all fundamentally change once we move beyond the Euclidean, single-species setting. This also highlights a key distinction from Equivariant Flow Matching [2] (EFM). EFM, still in the Euclidean case, builds on the invariance-preserving Continuous Normalizing Flow framework of [1] by training the flow using Flow Matching and incorporating equivariances into the optimal transport alignment. However, EFM does not analyze the invariances of the marginals or of the optimal velocity field, which are part of our theoretical development (Props. 4 and 9).
> 3. **Difference from crystal-based generative models.** Work on crystals [4,5,6] appears superficially closer, but the gap remains significant. They perform generative modeling (samples available, densities unknown), whereas we perform sampling (density known, samples unavailable). Although crystals may also involve multiple species and non-Euclidean manifolds, their representations and invariances are different from those of amorphous materials:  they are represented via lattices uniquely defined by Nigli reductions, leading to different invariance structures and modeling requirements.
>
> In summary, while the methodology is common with prior works, the theoretical extension to Riemannian settings and the substantially more complex physical systems we address constitute the core novelties of our paper, which are all needed to attack the problem of amorphous particle systems; these aspects are elaborated further in the global answer. We will update the manuscript to better reflect this point.

---

> > ### Author Response · Authors · 2025-11-26
> > **Answer to reviewer WnuX [2/2]**
> >
> > > The experiments are convincing, but arguably most of the improvement comes from the equivariance of the architecture [...] Have you tried out your method on different data/larger scales, and do you see similar improvements?
> >
> > The global answer also addresses this point. Prior works that sample particle systems using similar methodologies [1,2,3] operate at comparable dimensionalities (i.e., product between the number of particles and the ambient dimension), but focus on single-species, Euclidean-space systems. In contrast, the systems we consider (such as the inverse-power-law potential or the Kob-Andersen potential (App. F.4)) are already regarded as significantly more challenging, due to their multi-species structure, non-Euclidean geometry, and the inherent difficulty of sampling amorphous potentials [7]. We will improve the manuscript to better reflect this aspect. Because these amorphous-system potentials are known to be extremely hard to sample even for modest sizes [8], we did not push to larger scales here; doing so is a natural direction for future work, once the challenging setting we study is better understood.
> >
> > **References**
> >
> > [1] Kohler, J., Klein, L., & Noe, F. (2020). Equivariant Flows: Exact Likelihood Generative Learning for Symmetric Densities. In Proceedings of the 37th International Conference on Machine Learning (pp. 5361–5370). PMLR.
> >
> > [2] Klein, L., Krämer, A., & Noe, F. (2023). Equivariant flow matching. In Advances in Neural Information Processing Systems (pp. 59886–59910). Curran Associates, Inc..
> >
> > [3] Midgley, L., Stimper, V., Antorán, J., Mathieu, E., Schölkopf, B., & Hernandez-Lobato, J. (2023). SE(3) Equivariant Augmented Coupling Flows. In Advances in Neural Information Processing Systems (pp. 79200–79225). Curran Associates, Inc..
> >
> > [4] Rui Jiao, Wenbing Huang, Peĳia Lin, Jiaqi Han, Pin Chen, Yutong Lu, & Yang Liu (2023). Crystal Structure Prediction by Joint Equivariant Diffusion. In Thirty-seventh Conference on Neural Information Processing Systems.
> >
> > [5] Miller, B., Chen, R., Sriram, A., & Wood, B. (2024). FlowMM: Generating Materials with Riemannian Flow Matching. In Proceedings of the 41st International Conference on Machine Learning (pp. 35664–35686). PMLR.
> >
> > [6] Sriram, A., Miller, B., Chen, R., & Wood, B. (2024). FlowLLM: Flow Matching for Material Generation with Large Language Models as Base Distributions. In Advances in Neural Information Processing Systems (pp. 46025–46046). Curran Associates, Inc.
> >
> > [7] Berthier, L., & Reichman, D. (2023). Modern computational studies of the glass transition. Nature Reviews Physics, 5(2), 102–116.
> >
> > [8] Jung, G., Biroli, G., & Berthier, L. (2024). Normalizing flows as an enhanced sampling method for atomistic supercooled liquids. Machine Learning: Science and Technology, 5(3), 035053.

---

### Author Response · Authors · 2025-11-26
**General response [1/4]**

We thank all reviewers for their thoughtful and constructive feedback, as well as for their positive assessment of our work. Overall, the reviewers recognized the rigor of our theoretical framework and the effectiveness of our sampling methodology. In particular:

* Reviewer **WnuX** found the paper “well presented,” highlighted that the theoretical framework is “rigorous and thorough” and formalizes “intuitive claims,” and noted that the empirical evidence “clearly shows improvement” and is “very thorough.”
* Reviewer **73q7** emphasized that our method avoids issues encountered by competing approaches and noted that the “reweighting […] is shown to make a big improvement in performance.”
* Reviewer **kVyB** found the paper “well-structured and easy to follow.”
* Reviewer **NrV2** stated that the paper is “well written” and “quite readable,” addresses a topic that is “timely and of great interest,” and provides enough detail to make the results reproducible.

We address all concerns in detail below, and we have revised the manuscript accordingly. All changes are highlighted in blue in the updated version.

---

> ### Author Response · Authors · 2025-11-26
> **General response [2/4]**
>
> # 1. Novelty of the work
>
> Reviewers **WnuX** and **73q7** questioned the level of novelty in our theoretical and practical contributions. Reviewer **WnuX** noted that the theoretical framework “mostly formalises results that intuitively seem self-evident,” while Reviewer **73q7** pointed out that “many of these symmetries are closely discussed in other works.” We would like to clarify why our contributions are indeed novel and substantially distinct from prior work.
>
> As noted in our Related Works (Sec. 4), several papers have applied modern generative modeling ideas to particle systems. However, our setting is fundamentally different. Our work belongs to the line of research [1,2,3,4,5] that performs sampling of equilibrium distributions, not generative modeling. This distinction is essential: we do not have a dataset of samples but instead must sample from a known energy function, a task that is significantly more challenging than data-driven generative modeling.
>
> Compared to [1,2,3,4,5], the main novelty of our work is that we tackle substantially more complex physical systems: multi-species particle systems living on a non-Euclidean manifold and equipped with non-linear invariances. This has not been addressed before. This endeavor requires extending the mathematical framework of equivariant Continuous Normalizing Flows [1,2] to the non-Euclidean case and, beyond that, developing new theoretical guarantees (Prop. 4 and 9) ensuring that the optimal solutions respect the desired invariances. These results are specific to our RSI framework and are not available in prior work on Flow Matching [4,8].
>
> Other works have considered non-Euclidean, multi-species systems in the context of crystalline matter [6,7,8]. However, their setting differs from ours on several aspects: (i) they perform generative modeling, not sampling; (ii) the domain-specific representation of crystals (e.g., via periodic lattices) is different from amorphous configurations, yielding different invariances, architectures, and loss formulations; (iii) theoretical justification in these prior works is limited (e.g., [8] uses a custom conditional velocity field (Eq. 14) without grounding, treats marginal invariances only superficially, and handles species ambiguously).
>
> In summary, our work is novel because (a) we address the sampling problem rather than generative modeling, (b) we focus on non-crystalline, non-Euclidean multi-species systems with fundamentally different invariance structures, and (c) we provide a rigorous and new theoretical foundation enabling this setting. The closest work is [9], which also performs sampling on similar systems but uses a different generative modeling approach leveraging maximum likelihood trained continuous normalizing flows similar to [2].
>
> In the revised manuscript, we strengthen the exposition by emphasizing the conceptual difference between sampling and generative modeling, clarifying why the systems we study pose fundamentally new challenges, highlighting the novelty of our theoretical contributions, and making explicit the distinctions with crystal-based approaches at the level of task, theory, and methodology.

---

> > ### Author Response · Authors · 2025-11-26
> > **General response [3/4]**
> >
> > # 2. Numerical experiments: aims and challenges.
> >
> > Several reviewers expressed concern that our experiments may appear too simple at first glance. Reviewer **WnuX** asked for experiments on “different data/larger scales,” Reviewer **73q7** requested “bigger systems or ones with more motivated datasets” and “experimental data,” Reviewer **kVyB** asked for “larger-scale and real-world datasets,” and Reviewer **NrV2** noted the absence of “real data” such as “3-dimensional amorphous material.” We would like to clarify the nature and ambition of our experimental setting.
> >
> > As emphasized earlier, our work addresses the sampling problem rather than generative modeling. This distinction is crucial, and our experiments should be compared to prior sampling works on particle systems such as [1,2,3,4,5]. The typical benchmarks used in this sampling literature involve modest system sizes and classic Euclidean single-specie potentials:
> >
> > * [1]: 4 particles in 2D (toy potential)
> > * [2]: double-well with 2 or 4 particles in 2D; Lennard-Jones with 13 particles in 3D
> > * [3]: Lennard-Jones and monatomic water with up to 512 particles in 2D
> > * [4]: double-well with 4 particles in 3D; Lennard-Jones with 13 and 55 particles in 3D; 22-atom peptide
> > * [5]: double-well with 4 particles in 3D; Lennard-Jones with 13 particles in 3D; QM9 molecule with 19 atoms
> >
> > In comparison, our systems are at least as challenging, and structurally more complex. We study inverse-power-law potentials with 10 and 44 particles in 2D with two species, defined on a non-Euclidean manifold. Appendix F.4 further includes a three-species version of the Kob-Andersen potential, which is a widely used model [10]. These systems are canonical models of metallic glass formers, extensively used in theoretical and computational studies of amorphous materials [11,12]. In the context of glass models, it is well known that even systems containing tens of particles display the main features of complex glassy physics (slow relaxation, heterogeneous dynamics, resistance to crystallisation), and finite size effects are truly modest. It is also known that glassy physics is phenomenologically similar, and as challenging, in 2D and 3D systems, with experiments being performed in both 2 and 3 dimensions. Finally, both dimensions are notoriously hard to simulate, even at moderate system sizes [9,13]. In short, we would argue that the sampling problem for modest sizes in 2D glass models is a difficult and ambitious task.
> >
> > It is also important to distinguish our setting from works on crystalline materials [6,7,8]. While those systems also involve multiple species and non-Euclidean representations, no attempt at reweighting the configurations is made in these prior works. By contrast, Reviewer 73q7’s statement that “reweighting is shown to make a big improvement in performance”. For us, reweighting is not an optional feature but it is the very goal of the method, enabled precisely because we work with a known potential function. This makes our approach fundamentally different and not directly comparable to data-driven crystal generation.
> >
> > In summary, compared to prior sampling works, we handle systems that are different in nature but comparable (or more demanding) in difficulty while being directly relevant to physicists studying the glass transition; and comparisons with generative modeling works are not meaningful because the tasks, and available information differ.
> >
> > In the revised manuscript, we clarify that our objective is sampling from a potential rather than dataset-based generation, place our benchmarks in the context of the established literature on sampling particle systems, and clarify the relation (and differences) with crystal-based generative modeling works.

---

> ### Author Response · Authors · 2025-11-26
> **General response [4/4]**
>
> **References**
>
> [1] Jonas Köhler, Leon Klein, & Frank Noé. (2019). Equivariant Flows: sampling configurations for multi-body systems with symmetric energies.
>
> [2] Kohler, J., Klein, L., & Noe, F. (2020). Equivariant Flows: Exact Likelihood Generative Learning for Symmetric Densities. In Proceedings of the 37th International Conference on Machine Learning (pp. 5361–5370). PMLR.
>
> [3] Wirnsberger, P., Papamakarios, G., Ibarz, B., Racanière, S., Ballard, A., Pritzel, A., & Blundell, C. (2022). Normalizing flows for atomic solids. Machine Learning: Science and Technology, 3(2), 025009.
>
> [4] Klein, L., Krämer, A., & Noe, F. (2023). Equivariant flow matching. In Advances in Neural Information Processing Systems (pp. 59886–59910). Curran Associates, Inc..
>
> [5] Midgley, L., Stimper, V., Antorán, J., Mathieu, E., Schölkopf, B., & Hernandez-Lobato, J. (2023). SE(3) Equivariant Augmented Coupling Flows. In Advances in Neural Information Processing Systems (pp. 79200–79225). Curran Associates, Inc..
>
> [6] Sherry Yang, KwangHwan Cho, Amil Merchant, Pieter Abbeel, Dale Schuurmans, Igor Mordatch, & Ekin Cubuk (2023). Scalable Diffusion for Materials Generation. In NeurIPS 2023 AI for Science Workshop.
>
> [7] Rui Jiao, Wenbing Huang, Peĳia Lin, Jiaqi Han, Pin Chen, Yutong Lu, & Yang Liu (2023). Crystal Structure Prediction by Joint Equivariant Diffusion. In Thirty-seventh Conference on Neural Information Processing Systems.
>
> [8] Miller, B., Chen, R., Sriram, A., & Wood, B. (2024). FlowMM: Generating Materials with Riemannian Flow Matching. In Proceedings of the 41st International Conference on Machine Learning (pp. 35664–35686). PMLR.
>
> [9] Jung, G., Biroli, G., & Berthier, L. (2024). Normalizing flows as an enhanced sampling method for atomistic supercooled liquids. Machine Learning: Science and Technology, 5(3), 035053.
>
> [10] Kob, W.,  Andersen, H. C. (1994). Scaling Behavior in the $\beta$-Relaxation Regime of a Supercooled Lennard-Jones Mixture. Phys. Rev. Lett. 73, 1376.
>
> [11] Berthier, L., & Reichman, D. (2023). Modern computational studies of the glass transition. Nature Reviews Physics, 5(2), 102–116.
>
> [12] Berthier, L. & Biroli, G. (2011).  Theoretical perspective on the glass transition and amorphous materials. Reviews of Modern Physics, 83(2):587–645.
>
> [13] Jung, G., Ozawa, M., Biroli, G., Berthier, L. (2025). Numerical investigation of the equilibrium Kauzmann transition in a two-dimensional atomistic glass. arXiv:2507.03590.

---

### Meta-Review · Area_Chair_hd6Y · 2026-01-07

**Summary:**

This paper leverages techniques from generative modeling, in particular stochastic interpolant, and combines it with geometric treatments, to sample statistical distributions corresponding to the equilibrium distribution of amorphous particle system. Extrapolating diffusion-model-alike techniques from generative modeling tasks to sampling tasks is an active research frontier, and the main innovation of this paper is to focus on a specific non-Euclidean setup that incorporates periodic boundary conditions and the symmetries of multi-component particle systems. Some of the reviewers and I appreciate that this is an interesting setup. However, there are also major concerns about technical novelty, lack of important references, and weak experiments. Unfortunately, these concerns seem to remain largely unresolved after the rebuttal discussion. Recognizing the potential of the idea, I encourage the authors to take the discussions into consideration and re-submit a revised version.

**Reviewer Concerns:**

Just a guess: they probably will not be fully convinced.

**Reviewer Scores:**

Just a guess: they probably will not increase by too much.

---

### Decision · Program_Chairs · 2026-01-26

Reject